# Efficient Sliced Wasserstein Distance Computation via Adaptive Bayesian Optimization

**Manish Acharya and David Hyde**
Department of Computer Science
Vanderbilt University
Nashville, TN , USA
{manish.acharya,david.hyde.1}@vanderbilt.edu

## Abstract

The sliced Wasserstein distance (SW) reduces optimal transport on $\mathbb{R}^d$ to a sum of one-dimensional projections, and thanks to this efficiency, it is widely used in geometry, generative modeling, and registration tasks. Recent work shows that quasi-Monte Carlo constructions for computing SW (QSW) yield direction sets with excellent approximation error. This paper presents an alternate, novel approach: learning directions with Bayesian optimization (BO), particularly in settings where SW appears inside an optimization loop (e.g., gradient flows). We introduce a family of drop-in selectors for projection directions: **BOSW**, a one-shot BO scheme on the unit sphere; **RBOSW**, a periodic-refresh variant; **ABOSW**, an adaptive hybrid that seeds from competitive QSW sets and performs a few lightweight BO refinements; and **ARBOSW**, a restarted hybrid that periodically relearns directions during optimization. Our BO approaches can be composed with QSW and its variants (demonstrated by ABOSW/ARBOSW) and require no changes to downstream losses or gradients. We provide numerical experiments where our methods achieve state-of-the-art performance, and on the experimental suite of the original QSW paper, we find that ABOSW and ARBOSW can achieve convergence comparable to the best QSW variants with modest runtime overhead. We release code with fixed seeds and configurations to support faithful replication (see supplementary material).

## 1 Introduction

Optimal transport (OT) is a mathematical framework for measuring distances between probability measures (Villani et al., 2008). OT has seen increasing interest from the learning community in recent years since, for example, collections of sample data can be interpreted as empirical distributions. In particular, the Wasserstein distance is a popular metric for OT tasks in the learning literature (compared to, e.g., Kullback-Lieibler (KL) divergence, which is not symmetric, nor does it satisfy the triangle inequality) (Solomon et al., 2014; Montavon et al., 2016; Kolouri et al., 2017).

However, evaluating the conventional Wasserstein distance (WD) is computationally prohibitive ($\mathcal{O}(n^3 \log n)$ in time and $\mathcal{O}(n^2)$ in space (Nguyen et al., 2023a)), particularly for higher-dimensional measures, where computing Wasserstein distances can become the bottleneck of an application (Kolouri et al., 2019). The sliced Wasserstein (SW) distance (Bonneel et al., 2015a) offers a scalable alternative to the classical WD by averaging one-dimensional Wasserstein costs over projections on the unit sphere, giving $\mathcal{O}(n \log n)$ time and $\mathcal{O}(n)$ memory for discrete measures via sorting, and enabling its use across geometry, imaging, and generative modeling tasks (Peyré & Cuturi, 2019; Bonneel et al., 2015a)[1]. Formally, the SW between $\mu, \nu \in \mathcal{P}_p(\mathbb{R}^d)$ ($p \geq 1$) is

$$\mathrm{SW}_p^p(\mu,\nu) = \mathbb{E}_{\theta \sim \mathcal{U}(\mathbb{S}^{d-1})} \left[ W_p^p(\theta_\sharp \mu, \theta_\sharp \nu) \right].  \tag{1}$$

---

[1]Note that Bonnotte (2013) showed SW is equivalent to WD for compactly supported measures.

Here, $d$ is the ambient dimension, and $p \geq 1$ is the order of the Wasserstein cost. $\mathcal{P}_p(\mathbb{R}^d)$ denotes the set of Borel probability measures on $\mathbb{R}^d$ with finite $p$-th moment, i.e., $\int \|x\|^p \, d\mu(x) < \infty$. $\mathbb{S}^{d-1}$ is the unit sphere, and $\mathcal{U}(\mathbb{S}^{d-1})$ is the uniform distribution on it. For a direction $\theta \in \mathbb{S}^{d-1}$, the pushforward $\theta_\sharp \mu$ is the distribution of the scalar projection $\theta^\top X$ when $X \sim \mu$ (i.e., under $x \mapsto \theta^\top x$). $W_p(\cdot, \cdot)$ is the $p$-Wasserstein distance on $\mathbb{R}$; we average its $p$-th power over directions, i.e., $\mathrm{SW}_p^p(\mu, \nu) = \mathbb{E}_\theta[W_p^p(\theta_\sharp \mu, \theta_\sharp \nu)]$. Equivalently, one may write $\mathrm{SW}_p(\mu, \nu) = (\mathbb{E}_\theta[W_p^p(\theta_\sharp \mu, \theta_\sharp \nu)])^{1/p}$.

Of course, in practice, the expectation—which integrates over all directions—must be estimated using a finite set of directions. Given $L$ directions ("slices") $\Theta_L = \{\theta_\ell\}_{\ell=1}^L$, we estimate

$$\widehat{\mathrm{SW}}_p^p(\mu, \nu; \Theta_L) = \frac{1}{L} \sum_{\ell=1}^L W_p^p((\theta_\ell)_\sharp \mu, \, (\theta_\ell)_\sharp \nu). \tag{2}$$

For a fixed budget of $L$ slices, the quality of a SW estimate is therefore determined by the set $\Theta_L \subset \mathbb{S}^{d-1}$. For notational convenience, we denote $f(\theta; \mu, \nu) := W_p^p(\theta_\sharp \mu, \, \theta_\sharp \nu)$.

In the absence of any information, a natural choice for selecting the $\Theta_L$ is random Monte Carlo (MC) sampling, which famously suffers from a slow $\mathcal{O}(L^{-1/2})$ error convergence rate (Nadjahi et al., 2020). To mitigate this issue, recent work has applied quasi–Monte Carlo (QMC) sampling techniques to estimate SW (Nguyen et al., 2024a). However, QMC methods still sample pseudo-randomly, merely using data-independent heuristics to generate sample sets with more uniform coverage of the space being sampled (i.e., lower discrepancy) (Niederreiter, 1992; Dick & Pillichshammer, 2010). Randomized counterparts (RQSW) restore unbiasedness for stochastic optimization while retaining uniformity (Nguyen et al., 2024a).

However, in estimating SW, there is *not* an absence of information; the motivating idea of our paper is that when computing SW, *every slice yields additional information* that can inform the selection of further slices. For instance, if two nearby $\theta$ have substantially different values of $f(\theta; \mu, \nu)$, one may wish to sample additional slices between those two slices. Put another way, given a fixed budget of slices (especially for small $L$), the goal of selecting slices in SW should not be uniform coverage, but rather to maximize coverage where most of the "signal" of SW is determined.

Based on this insight, the present paper develops a novel family of algorithms for estimating SW, learning to select projection directions via Bayesian Optimization (BO), which proves to be especially effective in settings where SW is called inside an optimization loop (e.g., gradient-flow-style updates). BO has become a popular, sample-efficient strategy for selecting informative queries under tight budgets across hyperparameter tuning, experimental design, and robotics (Shahriari et al., 2016; Snoek et al., 2012; Garnett, 2023; Vardhan et al., 2024). Unlike QSW, which provides task-agnostic uniform coverage of the sphere, BO can exploit structure in the projection landscape to prioritize informative directions and adapt them as the task evolves. Our thesis is that when the objective repeatedly queries SW on evolving distributions, a small set of task-adapted directions can accelerate convergence without altering downstream losses or gradients—and that BO provides a simple, black-box way to pick them.

Of course, BO is not the only possibility for adaptive refinement of SW estimates. For instance, control variates such as the up/low method of Nguyen & Ho (2024) or spherical harmonics as in Leluc et al. (2024) have been shown to improve MC convergence in the context of SW. Repulsive point processes for MC have also been applied to SW (Petrovic et al., 2025), showing potential advantages in higher dimensions. Random-path projecting directions (Nguyen et al., 2024b) and Markovian SW (Nguyen et al., 2023b) both seek to select informative directions and show improved performance over baseline MC SW. Nonetheless, a recent survey on sampling for sliced OT (Sisouk et al., 2025) confirms BO remains unexplored in this area.

Our paper ultimately introduces and evaluates four drop-in, BO-based direction selectors: **BOSW**: a one-shot BO search on $\mathbb{S}^{d-1}$ to pick $L$ directions; **RBOSW**: a periodic-refresh variant that reuses BO for light retuning during optimization; **ABOSW**: an adaptive hybrid that seeds from strong QSW sets and applies a few lightweight BO refinements and **ARBOSW**: a restarted hybrid that periodically relearns directions (fresh BO) while retaining QSW seeding.

We note that our idea of leveraging BO can be complementary to QSW approaches: for instance, combining QSW and BO yields our ABOSW and ARBOSW methods, which often achieve the best

performance in our numerical results. RQSW-style randomization can still be layered when unbiased stochastic gradients are required (under the constructions proposed in Nguyen et al. (2024a)).

The contributions of our work include:

1. We provide a family of simple, plug-and-play selectors for SW projections driven by BO. In appropriate settings, these methods achieve state-of-the-art performance (measured by error vs. number of iterations) for estimating SW with a *finite* number of slices (BO makes estimators biased and prevents convergence to the true, unbiased SW value in the limit).

2. We integrate these methods into the QSW/RQSW pipeline without changing downstream losses/gradients.

3. On the QSW paper's test suite (Nguyen et al., 2024a), our hybrid variants (ABOSW/ARBOSW) show convergence competitive with the best QSW baselines at modest overhead; we follow their reporting protocol for one-to-one comparability.

## 2 BACKGROUND

**Wasserstein and sliced Wasserstein.** Consider a cost function or metric $c : \mathbb{R}^d \times \mathbb{R}^d \to \mathbb{R}_+$, and let $\mathcal{P}_c(\mathbb{R}^d)$ be the set of probability measures on $\mathbb{R}^d$ where there exists an $x_0$ such that $\int_{\mathbb{R}^d} c(x, x_0) d\mu(x) < \infty$. Then, given two measures $\mu, \nu \in \mathcal{P}_c(\mathbb{R}^d)$, the $p$-Wasserstein distance ($p \geq 1$) between $\mu$ and $\nu$ is defined by

$$W_{c,p}^p(\mu, \nu) = \min_{\pi \in \Pi(\mu,\nu)} \int c(x,y)^p \, \mathrm{d}\pi(x,y),$$

where $\Pi(\mu, \nu)$ denotes couplings with marginals $\mu, \nu$ (Nguyen, 2025; Peyré & Cuturi, 2019). The standard choice of $c(x, y)$ is $||x - y||_p$, in which case the $c$ is conventionally omitted from subscripts.

Computing $W_{c,p}$ in high dimension can be costly (cf. Section 1)). The sliced Wasserstein (SW) distance mitigates this by averaging one-dimensional Wasserstein costs over directions on the unit sphere (see Equation 1). Following the notation of Nguyen (2025), the *one-dimensional* Wasserstein-$p$ distance between $\mu, \nu \in \mathcal{P}_c(\mathbb{R})$ is

$$W_{c,p}^p(\mu, \nu) = \int_{\mathbb{R}} c(x, F_\nu^{-1} \circ F_\mu(x)) d\mu(x),$$

where $F_\mu$ is the cumulative distribution function (CDF) of $\mu$ and $F_\nu^{-1}$ is the generalized inverse CDF of $\nu$:

$$F_\nu^{-1}(x) = \inf\{y \in \mathbb{R} | x \leq F_\nu(y)\}.$$

Nguyen (2025) elucidate (see their Remark 2.9) that the computational complexity of evaluating $W_{c,p}^p$ in the one-dimensional case is $\mathcal{O}(n \log n)$ in time and $\mathcal{O}(n)$ in space.

As discussed in Section 1, we use the finite-direction estimator in Equation 2 with a set of slices $\Theta_L = \{\theta_\ell\}_{\ell=1}^L$; With i.i.d. Monte Carlo (MC) directions, the root mean square error scales as $\mathcal{O}(L^{-1/2})$.

**Quasi–Monte Carlo Methods.** Quasi–Monte Carlo (QMC) methods replace uniform random draws by low-discrepancy point sets (Niederreiter, 1992; Dick & Pillichshammer, 2010), improving integration error for sufficiently smooth integrands (e.g., a convergence rate of $\mathcal{O}(L^{-1}(\log L)^d)$ vs. $\mathcal{O}(L^{-1/2})$ for MC (Caflisch, 1998)). The idea of QMC for integrals dates back at least 50 years (Zaremba, 1968). In either MC or QMC methods, an integral is estimated as a discrete average of function evaluations (cf. Equation 2):

$$\int f(\mathbf{x}) d\mathbf{x} \approx \frac{1}{L} \sum_{i=1}^{L} f(\mathbf{x}_i).$$

The distinction of QMC is rather than uniformly randomly sampling points from the domain of $f$, one uses quasi-random (i.e., low-discrepancy) sequences of points. For instance, in one dimension, a van der Corput sequence (van der Corput, 1935) takes the positive natural numbers, expresses

them in a base (commonly base 2), reverses the digits, moves the digits after the radix point, and translates back into decimal (for instance, $5 \rightarrow 101_2 \rightarrow 101_2 \rightarrow 0.101_2 \rightarrow 0.625$). This scheme is repeated using different bases in different dimensions to form Halton sequences (Halton, 1964), a common multi-dimensional quasi-random scheme. Sobol sequences (Sobol, 1967) are related, but notably, they use only base 2, enabling fast computational arithmetic (Pharr et al., 2023).

**QSW and RQSW.** QSW instantiates QMC on $\mathbb{S}^2$ via (i) equal-area transforms of Sobol points, (ii) Gaussianized Sobol points (Ökten & Göncü, 2011) projected to the sphere, and (iii) structured deterministic sets such as spiral sequences and energy/distance-optimized designs; it also studies randomized counterparts (RQSW) that either scramble the net or apply random rotations, which restore unbiasedness while retaining uniform coverage (Nguyen et al., 2024a). Since these algorithms have state-of-the-art performance, we evaluate our methods against the same experimental protocols (datasets, metrics, and reporting) as Nguyen et al. (2024a) for fair comparison.

**Bayesian optimization (BO).** BO is a state-of-the-art sample-efficient strategy for optimizing expensive, black-box functions (Shahriari et al., 2016; Snoek et al., 2012; Garnett, 2023; Vardhan et al., 2024). BO works by first fitting a surrogate model $\mathcal{M}$ to a set of seed observations $\mathcal{D}_n = \{(x_i, f(x_i))\}_{i=1}^n$. With $\mathcal{M}$ as a Gaussian process (GP), for example, the posterior predictive distribution at a *new* point $x$ is normally distributed:

$$f(x) \mid \mathcal{D}_n \sim \mathcal{N}(\mu(x), \sigma^2(x)),$$

where $\mu$ and $\sigma^2$ are posterior mean and variance. $\sigma^2$ depends on a kernel (covariance) function $k$. For details on how the GP hyperparameters are fit to the data, we refer to the excellent tutorial and visualizations of Duque (2023).

To determine the next point to sample, BO optimizes an acquisition function $\alpha(x)$ and adds the resulting $x$ to $\mathcal{D}_n$. A popular choice of acquisition function is the upper confidence bound (UCB), which, for a GP surrogate, takes the form

$$\alpha_{\text{UCB}}(x; \beta) = \mu(x) + \beta\sigma(x),$$

where $\beta$ is a scalar hyperparameter that balances mean (exploitation) and variance (exploration) terms. The exploitation term favors candidate points that look promising based on the GP prior, whereas the exploration term pushes BO to select points from undersampled regions where little is known (large predictive variance).

**Problem view for this work.** Many learning tasks repeatedly evaluate sliced Wasserstein distances between an evolving distribution and a fixed target, e.g., gradient flows, registration, and neural network training. In such settings, the projection directions act purely as a numerical estimator and can therefore be adapted to the task at hand without changing the underlying optimization objective. Our BO-based methods learn such task-adapted direction sets: BOSW and ABOSW learn a fixed set once, while RBOSW and ARBOSW periodically refine the set as the distribution evolves. All variants drop into existing SW pipelines, e.g., those from Nguyen et al. (2024a), without modifying downstream losses or gradients.

## 3 METHODS

### 3.1 BAYESIAN OPTIMIZATION ON THE SPHERE

We treat $f(\theta; \mu, \nu)$ from Equation 2 as a black-box function on $\mathbb{S}^{d-1}$ and fit a GP surrogate with covariance (kernel) $k(\cdot, \cdot)$. We use the *angular RBF* kernel, i.e., a Gaussian of the spherical geodesic distance:

$$k(\theta, \theta') = \exp\left(-\tfrac{1}{2}\left(\tfrac{d_{\mathbb{S}}(\theta, \theta')}{\ell}\right)^2\right), \qquad d_{\mathbb{S}}(\theta, \theta') = \arccos\langle\theta, \theta'\rangle, \tag{3}$$

with lengthscale $\ell > 0$ (set by a median heuristic over pairwise geodesic distances among the currently evaluated directions). This isotropic (zonal) kernel on spheres is standard (Schoenberg, 1942; Borovitskiy et al., 2020; Rasmussen & Williams, 2006).

Given a budget of $L$ slices $\{\theta_i\}_{i=1}^L$, the goal of Bayesian optimization in the context of SW is to select the slices to maximize information about the integral in Equation 2. This is a sequential

experimental design problem—selecting which projection directions $\theta \in S^{d-1}$ to evaluate next, given previous observations.

At iteration $t$, with data $\mathcal{D}_t = \{(\theta_i, f(\theta_i))\}_{i=1}^{n_t}$ (so $n_t = |\mathcal{D}_t|$), the GP posterior $(\mu_t, \sigma_t)$ defines an acquisition $\alpha_t(\theta)$; by default we use UCB with $\beta = 0.7$ (Srinivas et al., 2010; Jones et al., 1998; Shahriari et al., 2016). New directions are proposed by maximizing $\alpha_t$ over a candidate pool of size $n_c$ sampled uniformly on $\mathbb{S}^{d-1}$ (we use $n_c = 4096$ unless stated), selecting a small batch $b$ (default $b = 5$). Candidates are normalized to unit norm and we suppress near-duplicates by dropping any proposal whose cosine similarity to the current set exceeds 0.98. These small, parallelizable rounds keep the BO overhead modest relative to SW evaluation; per round, scoring costs $\mathcal{O}(n_c \, n_t)$ kernel evaluations and adds only $b$ new $f$-evaluations.

We performed ablations to choose the models and parameters listed here; see Appendix C. Furthermore, we discuss the theoretical guarantees on using BO for SW in Appendix D.

### 3.2 Four Variations of Bayesian Optimization for Sliced Wasserstein

**BOSW: one-shot learned directions**   BOSW performs a single BO run to select $L$ directions, starting from a small random initialization and iteratively expanding $\Theta_L$ until full. The learned set remains fixed during downstream optimization. In terms of computational cost, each BO round requires $\mathcal{O}(n_c n_t)$ kernel evaluations, which is modest for $L$ in the hundreds. We note that the task-adapted directions BOSW learns do not comprise an unbiased quadrature rule for the spherical average. We include BOSW in the approximation error plot (see Section 4.2) only for completeness and do not expect it to excel there; for unbiased estimation we rely on (R)QSW. We omit RBOSW/ABOSW/ARBOSW from that plot since their refresh/seeding mechanisms target optimization-in-the-loop rather than a one-off integral.

**RBOSW: periodic refresh**   RBOSW re-runs the BOSW selection process every $R$ optimization steps using updated source and target distributions (i.e., the current pair of distributions whose distance is being measured with SW at step $t$). Between refreshes, the selected directions remain fixed. This adaptation can track evolving geometry without restarting the main optimization loop. We note that "refresh" here means re-running BOSW from scratch on the current data without carrying forward the previous surrogate model.

**ABOSW: QSW-seeded, lightweight refinement**   ABOSW is a variant that uses *adaptive* BO: standard GP-based BO (UCB) whose proposals are *conditioned on the slice values already observed*. ABOSW is adaptive to the task at initialization (QSW seed → a few BO refinements), but not time-adaptive; the set is refined once and then fixed.

ABOSW begins with a strong QSW set (e.g., spiral/Coulomb) and performs a few BO rounds to lightly adjust it:

1. **Initialize the GP** by evaluating $f(\theta)$ on the QSW seed $\Theta_L$; these evaluations form the initial dataset $\mathcal{D}_0$.

2. **Run a small number of BO rounds** (in our experiments, we run $r = 2$ rounds with mini-batch size $b = 5$; thus at most $br \le 10$ directions change when $L = 100$, i.e., $\le 10\%$ of the set). At each round: sample a candidate pool of size $n_c = 4096$ uniformly on $\mathbb{S}^{d-1}$, score by UCB (default $\beta = 0.7$), suppress near-duplicates (cosine similarity $> 0.98$), pick $b$ proposals, and replace the $b$ worst directions in $\Theta_L$.

3. **Return the refined set** $\Theta_L$; the overhead is small since only $br$ new directions are evaluated (mere seconds of compute, in our experiments).

**ARBOSW: restarted hybrid**   ARBOSW periodically restarts the ABOSW process: at fixed intervals, it re-seeds from a QSW set and runs the short BO refinement routine on current data. The key distinction from RBOSW is that each restart begins from a QSW initialization rather than the previous BO state, enabling both periodic adaptation and consistent seeding.

**Compatibility with randomized estimators**   When unbiased stochastic gradients are needed, randomized QSW remains appropriate (Nguyen et al., 2024a). Our BO-based selectors are orthogonal

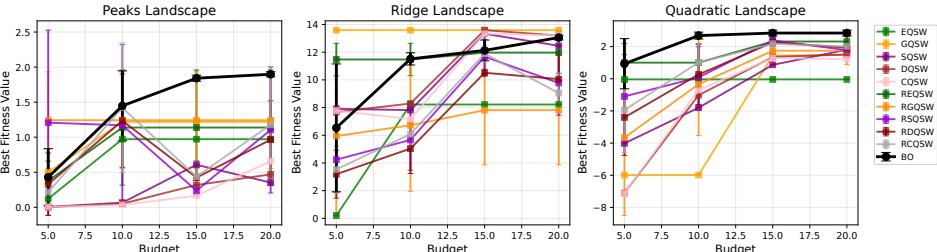

Figure 1: Synthetic projection-selection experiment on three fitness landscapes (higher is better). Bayesian optimization consistently and quickly finds the optimal projection directions. In the ridge example, one of the (pseudo-random) QSW methods happens to identify a useful slice early on; but BO clearly succeeds across all test cases.

to this choice: QSW (deterministic or randomized) provides the base set, and BO offers refinement or re-selection. We do not claim unbiasedness for BOSW-based sets (cf. Appendix D); evaluations follow the same deterministic vs. randomized protocols as Nguyen et al. (2024a).

### 3.3 COMPUTATIONAL OVERHEAD

The dominant cost of sliced Wasserstein evaluation is the $O(n \log n)$ sorting of $n$ projected points for each of the $L$ slices. BO-driven selectors add an additional cost of scoring $n_c$ candidate directions (we use $4096$) and evaluating a small batch of $b = 5$ new slices per refinement round, giving an overhead of $O(n_c n_t + b\, n \log n)$ relative to plain SW/QSW. In practice, this cost is modest compared to the core sorting operations: BOSW and ABOSW run within a few percent of QSW run-times, ARBOSW is roughly $1.5$–$2\times$ slower due to occasional refinements, and only RBOSW incurs substantial overhead due to its repeated refreshes. This pattern is consistent across all experiments, including point-cloud interpolation (Table 1), image translation, and autoencoding.

### 4 EXPERIMENTS

Aside from Section 4.1, our experiments match those run in Nguyen et al. (2024a): we use the same datasets, tasks, and reporting style, and we only "swap" the projection selector (ours vs. theirs). Across all tasks, we keep the optimizer, learning rates, and stopping criteria identical to Nguyen et al. (2024a) to ensure one-to-one comparability. We include the QSW families used in the paper, as well as vanilla Monte Carlo SW (denoted MCSW in our results). We note that in all our experiments, for methods with refreshes (RBOSW and ARBOSW), we use a refresh interval of $R = 25$, which we found to be a balance of speed and accuracy (see Appendix C).

### 4.1 SYNTHETIC PROJECTION-SELECTION EXPERIMENT

**Settings.** To verify that Bayesian optimization (BO) can adaptively identify high-quality projection directions, we construct synthetic "fitness landscapes" over the sphere where certain directions yield much higher rewards. We design three landscapes: (i) *Peaks*, with several local maxima and one dominant peak; (ii) *Ridge*, where projections aligned with $[1, 1, 1]$ achieve high value; and (iii) *Quadratic*, where a single target direction maximizes fitness. Each method is given a small evaluation budget $L \in \{5, 10, 15, 20\}$ and must return the best projection found. We compare BO against QSW baselines (EQSW, GQSW, SQSW, DQSW, CQSW) and their randomized counterparts (RQSW, RGQSW, RSQSW, RDQSW, RCQSW) (see Nguyen et al. (2024a) for details of each variant). Performance is measured by the mean best fitness value over 5 trials.

**Results.** Figure 1 summarizes the outcomes. On the *Peaks* landscape, BO steadily improves with budget and clearly outperforms all QSW variants at $L = 15$ and $20$ (best value $\approx 1.90$ versus $\approx 1.2$ for the strongest QSW). On the *Ridge* landscape, QSW methods that happen to align with the ridge (notably GQSW) perform well, but BO rapidly adapts to approach the optimum, surpassing

randomized variants at all budgets. On the *Quadratic* landscape, BO dominates: it already finds near-optimal projections by $L = 10$ (value 2.69 vs. $\leq 1.0$ for QSW/RQSW), and reaches the global optimum $\approx 2.84$ by $L = 15$, while QSW remains below 2.4. These results illustrate how BO leverages feedback to focus search on promising regions, whereas QSW's fixed designs (though effective for uniform coverage) cannot adapt to landscape structure. This synthetic experiment confirms BO's capability as an adaptive projection selector, motivating its use in sliced Wasserstein computations.

## 4.2 APPROXIMATION ERROR

**Setting.** We select four point clouds (indices 1–4; 2048 points; 3D) from the ShapeNet Core-55 (Chang et al., 2015) and use them as in Nguyen et al. (2024a). The population (ground truth) SW value for each pair (1–2, 1–3, 2–4, 3–4) is approximated by a high-budget MC estimate with $L = 100{,}000$. $L$ ranges from 10 to 10,000. We report the absolute error of: (i) MCSW, (ii) the QSW variants (GQSW, EQSW, SQSW, DQSW, CQSW), and (iii) our BOSW (one-shot BO). Stochastic methods (MCSW, BOSW) are averaged over five seeds; QSW variants are deterministic.

**Results.** Figure 2 shows absolute error versus $L$ across the four pairs. Unsurprisingly, QSW variants consistently yield lower errors than standard MC, with CQSW/DQSW (and SQSW/EQSW closely behind) approaching the $\approx 10^{-5}$ regime once $L \gtrsim 10^3$. In contrast, BOSW exhibits notably larger error across all $L$ and pairs, decreasing with $L$ but remaining above MC. This is expected: in this problem, the data are highly uniform, such that slices along each direction are roughly equally meaningful. Since BOSW learns task-adapted directions and is not designed to uniformly integrate the sphere, it does not perform competitively on approximation error for a task of this nature. Consequently, in subsequent sections, we focus on optimization-in-the-loop settings (e.g., gradient flows), where learned directions via BO are shown to help.

## 4.3 POINT-CLOUD INTERPOLATION

**Setting.** We evolve

$$\dot{Z}(t) \;=\; -n\,\nabla_{Z(t)}\big[\mathrm{SW}_2\big(P_{Z(t)}, P_Y\big)\big]$$

to interpolate between two point clouds $X$ and $Y$. Here, $P_X, P_Y$ are empirical distributions over $X$ and $Y$, and the curve starts at $Z(0) = X$ (and ends at $Y$). We use the same Euler scheme as Nguyen et al. (2024a): 500 iterations with step size 0.01, and evaluate the distance between $P_{Z(t)}$ and $P_Y$ using the 2-Wasserstein metric from POT (Flamary et al., 2021). We set $L = 100$ directions for all methods and report the mean $\pm$ standard deviation over thirty seeds at steps $t \in \{100, 200, 300, 400, 500\}$. We compare MCSW, the QSW family, the randomized QSW (RQSW) family, and our BO-driven selectors: BOSW (one-shot), RBOSW (periodic refresh), ABOSW (hybrid seed+refine), and ARBOSW (restarted hybrid).

**Results.** Table 1 reports $W_2$ ($\times 10^2$) and wall-clock time. As in Nguyen et al. (2024a), randomized QSW variants generally yield the shortest trajectories at later steps. However, our **ARBOSW**

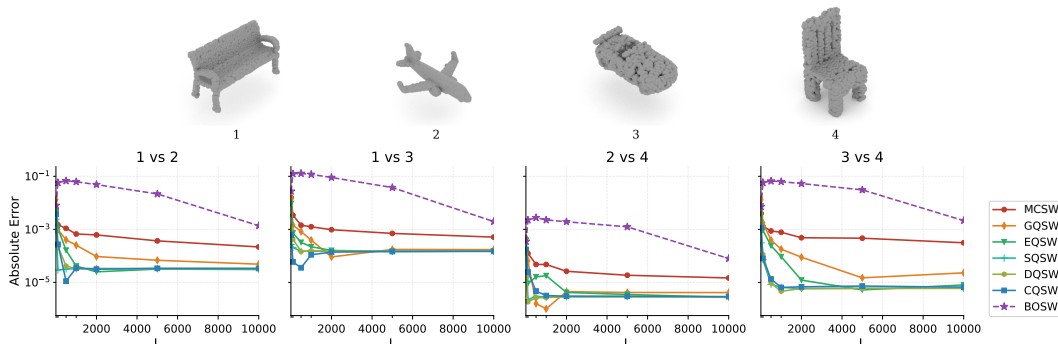

Figure 2: Approximation error for SW between empirical distributions over point clouds. BO does not provide an advantage compared to QSW methods, which excel on this type of problem that relies on uniformly sampling the sphere of possible directions.

achieves the *best* value at step 500, with a moderate overhead (6.9 s vs 3.8–4.2 s for QSW baselines). **RBOSW** is strongest early (best at steps 100 and 200) but incurs large cost due to periodic BO refreshes. The deterministic one-shot **BOSW** trails (consistent with the need for fresh directions as the flow evolves), while **ABOSW** performs similarly to the deterministic QSW group. Overall, BO hybrids provide competitive convergence without changing the optimization loop. For completeness, we also report results at a much smaller budget ($L = 10$ directions) in Appendix A.3. The trends are consistent: ARBOSW remains among the top-performing estimators, RBOSW dominates early steps but is slower, and BOSW/ABOSW align with their deterministic counterparts.

| Estimators | Step 100 (W$_2$ ↓) | Step 200 (W$_2$ ↓) | Step 300 (W$_2$ ↓) | Step 400 (W$_2$ ↓) | Step 500 (W$_2$ ↓) | Time (s↓) |
|---|---|---|---|---|---|---|
| MCSW | $5.749 \pm 0.079$ | $0.187 \pm 0.006$ | $0.031 \pm 0.002$ | $0.013 \pm 0.002$ | $0.006 \pm 0.002$ | 4.06 |
| GQSW | $6.082 \pm 0.0$ | $0.251 \pm 0.0$ | $0.078 \pm 0.0$ | $0.070 \pm 0.0$ | $0.069 \pm 0.0$ | 4.23 |
| EQSW | $5.408 \pm 0.0$ | $0.246 \pm 0.0$ | $0.083 \pm 0.0$ | $0.074 \pm 0.0$ | $0.072 \pm 0.0$ | 3.85 |
| SQSW | $5.707 \pm 0.0$ | $0.200 \pm 0.0$ | $0.085 \pm 0.0$ | $0.075 \pm 0.0$ | $0.074 \pm 0.0$ | 4.08 |
| DQSW | $5.771 \pm 0.0$ | $0.201 \pm 0.0$ | $0.075 \pm 0.0$ | $0.065 \pm 0.0$ | $0.064 \pm 0.0$ | 3.96 |
| CQSW | $5.603 \pm 0.0$ | $0.183 \pm 0.0$ | $0.078 \pm 0.0$ | $0.073 \pm 0.0$ | $0.071 \pm 0.0$ | 3.96 |
| RGQSW | $5.713 \pm 0.018$ | $0.182 \pm 0.004$ | $\mathbf{0.024 \pm 0.003}$ | $0.010 \pm 0.002$ | $0.004 \pm 0.001$ | 4.14 |
| RRGQSW | $5.735 \pm 0.035$ | $0.183 \pm 0.007$ | $0.029 \pm 0.003$ | $0.012 \pm 0.003$ | $0.005 \pm 0.002$ | 4.01 |
| REQSW | $5.705 \pm 0.023$ | $0.183 \pm 0.002$ | $0.029 \pm 0.002$ | $0.013 \pm 0.002$ | $0.007 \pm 0.002$ | 3.87 |
| RREQSW | $5.721 \pm 0.023$ | $0.183 \pm 0.003$ | $0.027 \pm 0.002$ | $0.011 \pm 0.002$ | $0.007 \pm 0.002$ | 3.80 |
| RSQSW | $5.710 \pm 0.001$ | $0.182 \pm 0.003$ | $0.026 \pm 0.002$ | $\mathbf{0.010 \pm 0.002}$ | $0.005 \pm 0.002$ | 3.95 |
| RDQSW | $5.711 \pm 0.004$ | $0.181 \pm 0.003$ | $0.027 \pm 0.002$ | $0.012 \pm 0.002$ | $0.005 \pm 0.001$ | 4.00 |
| RCQSW | $5.708 \pm 0.005$ | $0.181 \pm 0.002$ | $0.027 \pm 0.003$ | $0.011 \pm 0.003$ | $0.005 \pm 0.002$ | 3.95 |
| **BOSW** | $3.744 \pm 0.179$ | $0.217 \pm 0.040$ | $0.101 \pm 0.013$ | $0.091 \pm 0.010$ | $0.088 \pm 0.009$ | 3.81 |
| **RBOSW (ours)** | $\mathbf{2.213 \pm 0.055}$ | $\mathbf{0.083 \pm 0.008}$ | $0.047 \pm 0.005$ | $0.033 \pm 0.004$ | $0.025 \pm 0.003$ | 44.58 |
| **ABOSW (ours)** | $6.199 \pm 0.522$ | $0.298 \pm 0.084$ | $0.087 \pm 0.012$ | $0.077 \pm 0.009$ | $0.075 \pm 0.008$ | 3.95 |
| **ARBOSW (ours)** | $5.717 \pm 0.079$ | $0.186 \pm 0.004$ | $0.025 \pm 0.003$ | $0.012 \pm 0.001$ | $\mathbf{0.003 \pm 0.001}$ | 6.91 |

Table 1: Summary of Wasserstein-2 distances (multiplied by $10^2$) from 30 different runs (L=100).

## 4.4 IMAGE STYLE TRANSFER

**Setting.** RGB source and target image are treated as point clouds $X, Y \in \mathbb{R}^{n \times 3}$ ($n$ is the number of pixels) and are evolved along the same SW–driven curve as in Section 4.3. As in Nguyen et al. (2024a), we round RGB values to $\{0, \ldots, 255\}$ at the final Euler step, use $1{,}000$ iterations with step size 1, and set $L = 100$ projection directions for all methods; $W_2$ is computed with POT (Flamary et al., 2021). We visualize MCSW, **RCQSW**, and our **RBOSW/ARBOSW**. We report RCQSW as the representative RQSW baseline because the QSW study found it consistently among the top performers. Randomized variants behave very similarly overall; see Appendix A.6 for full results.

**Results.** Figure 3 shows a representative transfer. Both RCQSW and our BO hybrids clearly outperform vanilla MCSW in terms of final $W_2$ (printed above each panel) and visual fidelity. For ease of exposition, we keep **RCQSW** as the representative randomized baseline in the main text, mirroring the recommendation of Nguyen et al. (2024a); see Appendix A.4 for full comparisons. We note, for transparency, that other randomized variants can be marginally stronger on some instances; in particular, **RGQSW** achieves the lowest final $W_2$ on our example at both $L$=10 and $L$=100. However, the randomized family behaves very similarly overall, and RCQSW remains competitive. Our **ARBOSW** matches RCQSW on this example while preserving contrast and texture; **RBOSW** improves over SW but typically trails RCQSW/ARBOSW, consistent with its lighter periodic refresh. Thus, while not as clear of an accuracy advantage as seen in Sections 4.3 and 4.5, ARBOSW is at least competitive with the state-of-the-art for this example.

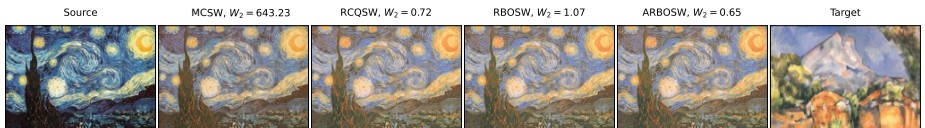

Figure 3: Image style transfer with $L$=100 and 1,000 iterations.

| Approximation | Epoch 100 | | Epoch 200 | | Epoch 400 | |
|---|---|---|---|---|---|---|
| | SW$_2$($\downarrow$) | W$_2$($\downarrow$) | SW$_2$($\downarrow$) | W$_2$($\downarrow$) | SW$_2$($\downarrow$) | W$_2$($\downarrow$) |
| MCSW | $2.25 \pm 0.06$ | $10.58 \pm 0.12$ | $2.11 \pm 0.04$ | $9.92 \pm 0.08$ | $1.94 \pm 0.06$ | $9.21 \pm 0.06$ |
| GQSW | $11.17 \pm 0.07$ | $32.58 \pm 0.06$ | $11.75 \pm 0.07$ | $33.27 \pm 0.09$ | $14.82 \pm 0.02$ | $37.99 \pm 0.05$ |
| EQSW | $2.25 \pm 0.02$ | $10.57 \pm 0.02$ | $2.05 \pm 0.02$ | $9.84 \pm 0.07$ | $1.90 \pm 0.04$ | $9.20 \pm 0.07$ |
| SQSW | $2.25 \pm 0.01$ | $10.57 \pm 0.03$ | $2.08 \pm 0.01$ | $9.90 \pm 0.04$ | $1.90 \pm 0.02$ | $9.17 \pm 0.05$ |
| DQSW | $2.24 \pm 0.07$ | $10.58 \pm 0.05$ | $2.06 \pm 0.04$ | $9.83 \pm 0.01$ | $1.86 \pm 0.02$ | $9.12 \pm 0.07$ |
| CQSW | $2.22 \pm 0.02$ | $10.54 \pm 0.02$ | $2.05 \pm 0.06$ | $9.81 \pm 0.04$ | $1.84 \pm 0.05$ | $9.06 \pm 0.02$ |
| RGQSW | $2.25 \pm 0.02$ | $10.57 \pm 0.01$ | $2.09 \pm 0.03$ | $9.92 \pm 0.01$ | $1.94 \pm 0.02$ | $9.18 \pm 0.02$ |
| RRGQSW | $2.23 \pm 0.01$ | $10.51 \pm 0.04$ | $2.06 \pm 0.05$ | $9.84 \pm 0.06$ | $1.88 \pm 0.09$ | $9.16 \pm 0.11$ |
| REQSW | $2.24 \pm 0.04$ | $10.53 \pm 0.04$ | $2.08 \pm 0.04$ | $9.90 \pm 0.08$ | $1.89 \pm 0.04$ | $9.17 \pm 0.06$ |
| RREQSW | $2.21 \pm 0.04$ | $10.50 \pm 0.04$ | $2.03 \pm 0.02$ | $9.83 \pm 0.02$ | $1.88 \pm 0.05$ | $9.15 \pm 0.06$ |
| RSQSW | $2.22 \pm 0.05$ | $10.53 \pm 0.01$ | $2.04 \pm 0.06$ | $9.82 \pm 0.06$ | $1.85 \pm 0.05$ | $9.12 \pm 0.02$ |
| RDQSW | $2.21 \pm 0.03$ | $10.50 \pm 0.02$ | $2.03 \pm 0.04$ | $9.82 \pm 0.04$ | $1.86 \pm 0.03$ | $9.12 \pm 0.02$ |
| RCQSW | $2.22 \pm 0.03$ | $10.50 \pm 0.05$ | $2.03 \pm 0.02$ | $9.82 \pm 0.03$ | $1.85 \pm 0.06$ | $9.12 \pm 0.03$ |
| BOSW | $2.20 \pm 0.01$ | $10.34 \pm 0.02$ | $2.02 \pm 0.04$ | $9.78 \pm 0.03$ | $\mathbf{1.80 \pm 0.01}$ | $\mathbf{9.01 \pm 0.02}$ |
| RBOSW | $2.28 \pm 0.03$ | $10.54 \pm 0.04$ | $2.09 \pm 0.05$ | $9.84 \pm 0.05$ | $1.90 \pm 0.02$ | $9.10 \pm 0.03$ |
| ABOSW | $\mathbf{2.18 \pm 0.01}$ | $\mathbf{10.27 \pm 0.02}$ | $\mathbf{2.01 \pm 0.03}$ | $\mathbf{9.76 \pm 0.02}$ | $1.81 \pm 0.02$ | $\mathbf{9.01 \pm 0.03}$ |
| ARBOSW | $2.21 \pm 0.01$ | $10.44 \pm 0.02$ | $2.04 \pm 0.04$ | $9.80 \pm 0.03$ | $1.85 \pm 0.02$ | $9.07 \pm 0.02$ |

Table 2: Reconstruction losses on the autoencoder task from different approximations with $L = 100$. Losses are scaled by $10^2$ for ease of exposition.

## 4.5 DEEP POINT-CLOUD AUTOENCODER

**Setting.** Following Nguyen et al. (2023a) and Nguyen et al. (2024a), we train deep point-cloud autoencoders with SW on ShapeNet Core-55 (Chang et al., 2015). This amounts to optimizing an objective function:

$$\min_{\phi,\psi} \ \mathbb{E}_{X \sim \mu(X)} \big[ \mathrm{SW}_p \big( P_X, P_{g_\psi(f_\phi(X))} \big) \big],$$

where $\mu(X)$ is the data distribution, $f_\phi$ is a deep encoder, and $g_\psi$ is a deep decoder. Both $f_\phi$ and $g_\psi$ use a PointNet architecture (Qi et al., 2017). We approximate gradients using either MC, QSW, RQSW, or our BO-driven selectors, and train for $400$ epochs with SGD (learning rate $10^{-3}$, batch size $128$, momentum $0.9$, weight decay $5 \times 10^{-4}$). We set $L = 100$ directions for all methods. Evaluation is on the distinct ModelNet40 dataset (Wu et al., 2015); we report the mean reconstruction loss over three runs, measured against $W_2$ and SW$_2$ (which are estimated via 10,000 MC projections).

**Results.** Table 2 and Figure 4 show the reconstruction losses and qualitative outputs with $L = 100$. Consistent with prior work, CQSW is a strong deterministic baseline. However, our BO-based methods, particularly **ABOSW**, achieve the lowest reconstruction losses across epochs, even surpassing CQSW in both SW$_2$ and $W_2$ at the final epoch. BOSW also performs competitively, while the restart variants (RBOSW, ARBOSW) lag slightly behind their non-restart counterparts. This reversal compared to gradient flows highlights the different nature of dataset-level training: here, the distribution is large and stable, so maintaining consistent, high-quality projection sets (as in BOSW/ABOSW) is more effective than frequent refreshes, which can inject unnecessary variability. We emphasize that while methods like RGQSW, RRGQSW, and REQSW were top performers for image style transfer (Section 4.4), they are among the worst-performing methods in Table 2 after 400 epochs. Overall, BO-based selectors not only remain competitive but, in this setting, **outperform the previously recommended CQSW**, suggesting that task-adaptivity via BO can provide tangible benefits beyond the low-discrepancy sampling enabled by QSW. For clarity, we include MCSW, CQSW, and our BO variants in the main text; the full visual and quantitative comparisons with all QSW/RQSW variants are deferred to Appendix A.6.

## 5 CONCLUDING REMARKS

We proposed BOSW, RBOSW, ABOSW, and ARBOSW: Bayesian optimization–driven selectors for projection directions in sliced Wasserstein (SW) computations that complement quasi–Monte Carlo (QSW) designs. Our hybrids combine QSW's low-discrepancy coverage with BO's task adaptivity and drop in to existing SW-based optimization loops without changing losses or gradients. Across the QSW benchmark suite of Nguyen et al. (2024a), ABOSW and ARBOSW match the

best QSW baselines in convergence, RBOSW offers strong early-stage gains, and BOSW provides a simple deterministic alternative. Collectively, our methods perform **competitively** on the image style transfer task, and achieve **state-of-the-art** convergence for the point-cloud interpolation and deep point-cloud autoencoder tasks. By uniting deterministic geometric constructions with data-driven adaptivity, our work broadens the design space for SW direction sets and opens a path toward principled, learned projections in large-scale optimal transport applications.

Reflecting on our proposed algorithms, we remark on the work of Nguyen & Ho (2023), where the authors reweight slices (directions) for SW based on the slices' 1D costs, yielding a new slicing distribution. In the view of that work, our proposed nonparametric regression framework uses in- and out-of-sample weights—training directions and new proposed directions, respectively—to go beyond simply defining a set of projection directions and effectively define a new slicing distribution (or, alternatively, a data-dependent Radon transform (Bonneel et al., 2015b; Kolouri et al., 2019)). We note that while the importance sampling energy-based SW of Nguyen & Ho (2023) is not an unbiased estimator of SW for finite $L$, it *is* an unbiased estimator of a new metric it induces ("importance-weighted SW"). Thus, as future work, we are interested in further analyzing bias of and metrics induced by our BO-based methods in light of Nguyen & Ho (2023).

A common criticism of BO is its poor scaling with respect to the dimension $d$ of the data. However, recent literature has suggested several variants of BO that perform well in high dimensions (Kim et al., 2021; Shen & Kingsford, 2021; Binois & Wycoff, 2022; Jaquier & Rozo, 2020), and in an exciting development, Hvarfner et al. (2024) just found that even vanilla BO can perform excellently in high dimensions with a simple scaling of the GP lengthscale. Moreover, the GP surrogate, which is a primary bottleneck in BO, could potentially be replaced with a neural surrogate for performance that scales better with dimension (Li et al., 2023; Lim et al., 2021). Thus, overall, it is no longer fair to claim that BO is cursed by dimensionality. We would like to test our methods on higher-dimensional datasets in future work.

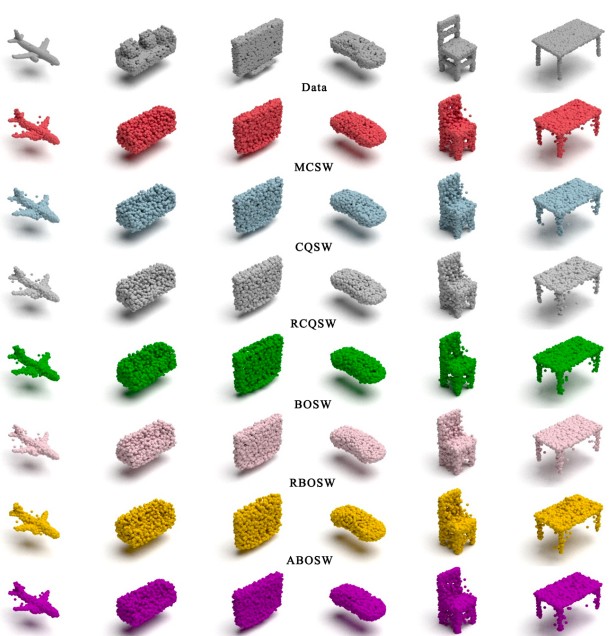

Figure 4: Reconstructed point clouds from the deep autoencoder experiment, with $L = 100$.

We remark that the present work is reminiscent of (adaptive) Bayesian quadrature (BQ) (Duvenaud, 2012; Kanagawa & Hennig, 2019; Briol et al., 2015; Osborne et al., 2012). However, BQ is designed to compute an integral itself, not select individual directions. In our setting, the integrand changes every step, so BQ would require refitting a GP and recomputing quadrature weights each time, for a total of $O(L^3)$ runtime; see Section A.2 for a short experiment. Thus, while BQ is infeasible to directly apply, it remains closely related to the ideas in our work.

Finally, we imagine several opportunities to improve our methods. Parallelism and GPU implementation, as explored by others in the field (Balandat et al., 2020; Knudde et al., 2017; Munawar et al., 2009), could significantly accelerate the runtime of our methods. Adding constraints or information to BO based on knowledge of the problem, as in Eriksson & Poloczek (2021); Gelbart (2015); Vardhan et al. (2023); Jaquier et al. (2022), would give our methods further advantage over data-blind techniques like QSW. Finally, recent variants of BO could offer higher-performance drop-in replacements for the BO functions in our paper (Visser et al., 2025; McLeod et al., 2018; Kawaguchi et al., 2015). We aim to explore these extensions in future work.

ACKNOWLEDGMENTS

This paper was inspired by a Ph.D. preliminary exam of a student advised by Soheil Kolouri, where the authors found that sliced Wasserstein is about approximating integrals accurately; and by the DARPA Design.R project, which introduced the authors to Bayesian optimization. D.H. acknowledges Peter Volgyesi for involving him in the Design.R efforts. We thank the SyBBURE Searle Undergraduate Research Program for providing funding and support throughout this research. This material is based upon work supported by the National Science Foundation under Grant No. 2442853.

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

APPENDIX

# A  ADDITIONAL EXPERIMENTAL DETAILS AND RESULTS

## A.1  SENSITIVITY TO KERNEL CHOICE AND ACQUISITION PARAMETERS

**Setup.**  We evaluate the robustness of our BO-based direction selectors by varying the Gaussian process kernel (RBF vs. Matérn–3/2) and the UCB acquisition parameter $\beta \in \{1, 2, 3\}$. We choose these kernels because they are the standard GP covariances used on the sphere (Borovitskiy et al., 2020). Experiments follow the approximation-error setting from Section 4.1 with $L \in \{100, 1000\}$, four ShapeNet point-cloud pairs, and five random seeds. We focus on BOSW (fixed direction set learned once) and ARBOSW (periodically refined set), which represent the two BO refinement regimes used in the paper. The other variants, ABOSW and RBOSW, use the same BO inner loop and exhibited identical kernel/acquisition behavior in preliminary tests, so we omit them for brevity.

**Results.**  Across all kernel and $\beta$ choices, both BOSW and ARBOSW show extremely small variation in approximation error. The standard deviation remains below $4 \times 10^{-5}$ for $L{=}100$ and below $8 \times 10^{-6}$ for $L{=}1000$, and the relative ranking of methods is unchanged. RBF and Matérn–3/2 kernels yield nearly identical results. These findings confirm that our BO-driven direction selection is highly robust to kernel and acquisition hyperparameters.

| Method | Kernel | $\beta$ | L=100 | L=1000 |
|---|---|---|---|---|
| BOSW | RBF | 1 | $(9.74{\times}10^{-5} \pm 4.10{\times}10^{-5})$ | $(2.20{\times}10^{-5} \pm 7.12{\times}10^{-6})$ |
| BOSW | RBF | 2 | $(9.74{\times}10^{-5} \pm 4.10{\times}10^{-5})$ | $(2.20{\times}10^{-5} \pm 7.12{\times}10^{-6})$ |
| BOSW | RBF | 3 | $(9.41{\times}10^{-5} \pm 4.16{\times}10^{-5})$ | $(2.31{\times}10^{-5} \pm 6.70{\times}10^{-6})$ |
| BOSW | Matérn–3/2 | 1 | $(9.62{\times}10^{-5} \pm 4.11{\times}10^{-5})$ | $(2.26{\times}10^{-5} \pm 6.81{\times}10^{-6})$ |
| BOSW | Matérn–3/2 | 2 | $(9.62{\times}10^{-5} \pm 4.11{\times}10^{-5})$ | $(2.26{\times}10^{-5} \pm 6.81{\times}10^{-6})$ |
| BOSW | Matérn–3/2 | 3 | $(9.62{\times}10^{-5} \pm 4.11{\times}10^{-5})$ | $(2.26{\times}10^{-5} \pm 6.81{\times}10^{-6})$ |
| ARBOSW | RBF | 1 | $(9.74{\times}10^{-5} \pm 4.10{\times}10^{-5})$ | $(2.20{\times}10^{-5} \pm 7.12{\times}10^{-6})$ |
| ARBOSW | RBF | 2 | $(9.74{\times}10^{-5} \pm 4.10{\times}10^{-5})$ | $(2.20{\times}10^{-5} \pm 7.12{\times}10^{-6})$ |
| ARBOSW | RBF | 3 | $(9.41{\times}10^{-5} \pm 4.16{\times}10^{-5})$ | $(2.31{\times}10^{-5} \pm 6.70{\times}10^{-6})$ |
| ARBOSW | Matérn–3/2 | 1 | $(9.74{\times}10^{-5} \pm 4.10{\times}10^{-5})$ | $(2.20{\times}10^{-5} \pm 7.12{\times}10^{-6})$ |
| ARBOSW | Matérn–3/2 | 2 | $(9.74{\times}10^{-5} \pm 4.10{\times}10^{-5})$ | $(2.20{\times}10^{-5} \pm 7.12{\times}10^{-6})$ |
| ARBOSW | Matérn–3/2 | 3 | $(9.41{\times}10^{-5} \pm 4.16{\times}10^{-5})$ | $(2.31{\times}10^{-5} \pm 6.70{\times}10^{-6})$ |

Table 3: Kernel and acquisition-parameter sensitivity (mean $\pm$ std over four ShapeNet pairs and five seeds).

## A.2  FEASIBILITY OF CLASSICAL BAYESIAN QUADRATURE (BQ)

Classical Bayesian Quadrature (BQ) is conceptually related but computationally impractical for sliced-Wasserstein projection selection: each iteration requires refitting a GP and recomputing quadrature weights, leading to overall $O(L^3)$ cost (see Section 5).

To illustrate this scaling, we benchmarked a standard adaptive BQ method (GP refit + UCB acquisition) against SW and our ABOSW method on synthetic point clouds. Table 4 reports the end-to-end runtime for generating all $L$ projection directions.

| Method | $L=50$ | $L=100$ | $L=200$ | $L=500$ | $L=1000$ | $L=2000$ | $L=5000$ |
|---|---|---|---|---|---|---|---|
| MCSW | 0.00 | 0.00 | 0.00 | 0.00 | 0.01 | 0.01 | 0.03 |
| ABOSW | 0.35 | 0.06 | 0.06 | 0.06 | 0.06 | 0.07 | 0.09 |
| BQ | 0.10 | 0.21 | 0.47 | 1.61 | 5.78 | 35.73 | 654.66 |

Table 4: Wall-clock runtime (seconds) for generating $L$ projection directions.

ABOSW shows a slightly higher cost at $L$=50 because its adaptive initialization performs more refinement steps when the budget is small; for larger budgets ($L \geq 100$), this cost is amortized and ABOSW switches quickly into its fast QSW-style fill-in stage, yielding a nearly flat runtime of 0.06–0.09s across all $L$.

In contrast, BQ grows from 0.47s at $L$=200 to 1.61s at $L$=500, 5.78s at $L$=1000, and over 650s at $L$=5000, fully consistent with its $O(L^3)$ scaling. These results further support our decision not to include classical BQ as a comparative baseline in the main experiments.

### A.3 POINT-CLOUD INTERPOLATION WITH $L = 10$ DIRECTIONS

| Estimators | Step 100 ($W_2 \downarrow$) | Step 200 ($W_2 \downarrow$) | Step 300 ($W_2 \downarrow$) | Step 400 ($W_2 \downarrow$) | Step 500 ($W_2 \downarrow$) | Time (s$\downarrow$) |
|---|---|---|---|---|---|---|
| MCSW | $5.719 \pm 0.113$ | $0.190 \pm 0.016$ | $0.041 \pm 0.001$ | $0.020 \pm 0.002$ | $0.011 \pm 0.001$ | **2.71** |
| GQSW | $9.207 \pm 0.000$ | $3.692 \pm 0.000$ | $2.461 \pm 0.000$ | $2.191 \pm 0.000$ | $2.119 \pm 0.000$ | 2.92 |
| EQSW | $4.045 \pm 0.000$ | $0.552 \pm 0.000$ | $0.490 \pm 0.000$ | $0.487 \pm 0.000$ | $0.482 \pm 0.000$ | 2.78 |
| SQSW | $6.323 \pm 0.000$ | $1.044 \pm 0.000$ | $0.582 \pm 0.000$ | $0.538 \pm 0.000$ | $0.534 \pm 0.000$ | 2.84 |
| DQSW | $5.897 \pm 0.000$ | $0.856 \pm 0.000$ | $0.612 \pm 0.000$ | $0.595 \pm 0.000$ | $0.594 \pm 0.000$ | 2.79 |
| CQSW | $5.574 \pm 0.000$ | $0.755 \pm 0.000$ | $0.592 \pm 0.000$ | $0.582 \pm 0.000$ | $0.582 \pm 0.000$ | 2.87 |
| RGQSW | $6.036 \pm 0.231$ | $0.183 \pm 0.022$ | $0.034 \pm 0.003$ | $0.017 \pm 0.003$ | $0.009 \pm 0.002$ | 2.84 |
| RRGQSW | $6.002 \pm 0.149$ | $0.215 \pm 0.025$ | $0.047 \pm 0.004$ | $0.029 \pm 0.003$ | $0.024 \pm 0.002$ | 2.78 |
| REQSW | $5.722 \pm 0.153$ | $0.189 \pm 0.008$ | $0.039 \pm 0.005$ | $0.019 \pm 0.004$ | $0.009 \pm 0.002$ | 2.80 |
| RREQSW | $5.832 \pm 0.018$ | $0.193 \pm 0.006$ | $0.038 \pm 0.002$ | $0.022 \pm 0.001$ | $0.015 \pm 0.000$ | 2.83 |
| RSQSW | $5.718 \pm 0.064$ | $0.187 \pm 0.008$ | $\mathbf{0.035 \pm 0.004}$ | $\mathbf{0.015 \pm 0.002}$ | $\mathbf{0.007 \pm 0.001}$ | 2.81 |
| RDQSW | $5.694 \pm 0.044$ | $\mathbf{0.182 \pm 0.020}$ | $0.033 \pm 0.002$ | $0.015 \pm 0.001$ | $0.007 \pm 0.002$ | 2.79 |
| RCQSW | $\mathbf{5.673 \pm 0.023}$ | $0.184 \pm 0.008$ | $0.038 \pm 0.005$ | $0.018 \pm 0.003$ | $0.008 \pm 0.002$ | 2.79 |
| **BOSW (ours)** | $4.833 \pm 0.863$ | $1.303 \pm 0.494$ | $0.967 \pm 0.375$ | $0.912 \pm 0.345$ | $0.900 \pm 0.333$ | 2.86 |
| **RBOSW (ours)** | $\mathbf{3.457 \pm 0.109}$ | $0.175 \pm 0.044$ | $0.082 \pm 0.030$ | $0.059 \pm 0.005$ | $0.047 \pm 0.004$ | 22.63 |
| **ABOSW (ours)** | $9.086 \pm 3.294$ | $3.005 \pm 1.499$ | $1.637 \pm 0.520$ | $1.297 \pm 0.205$ | $1.216 \pm 0.139$ | 3.09 |
| **ARBOSW (ours)** | $5.617 \pm 0.128$ | $0.207 \pm 0.015$ | $0.039 \pm 0.002$ | $0.017 \pm 0.003$ | $0.009 \pm 0.001$ | 3.09 |

Table 5: Summary of Wasserstein-2 distances (multiplied by $10^2$) from three different runs (L=10).

Table 5 reports results for point-cloud interpolation with a smaller projection budget of $L = 10$. As expected, variance across methods is larger under such limited directions, but the relative trends mirror those observed for $L = 100$: randomized QSW variants achieve fast convergence at later steps, while **ARBOSW** remains within the top four methods overall, offering competitive final accuracy. **RBOSW** continues to excel early in the trajectory but incurs a higher runtime due to periodic refreshes.

### A.4 COMPLETE IMAGE STYLE TRANSFER COMPARISONS

Figures 5 and 6 visualize the full method grid. Across both budgets, the randomized family performs very similarly; within that family **RGQSW** attains the *lowest* final $W_2$ on our example for *both* $L$=10 and $L$=100. This is consistent with the QSW observation that randomized designs are closely clustered in quality. Our BO hybrids remain competitive: **ARBOSW** is comparable to RCQSW visually and in $W_2$ at $L$=100, while **RBOSW** improves over SW but generally trails the strongest RQSW variants, reflecting its lighter refresh. These results support our choice to use RCQSW as the representative randomized baseline in the main figures while providing full transparency here.

### A.5 COLOR–TRANSLATION EXPERIMENTS AT $128 \times 128$

We additionally include a controlled color–translation study in which the source and target images differ only through a global additive RGB shift. This setting preserves the spatial structure of the image and isolates the effect of a distributional shift in color space. For a given $128 \times 128$ source image $X$, we construct the target image $\tilde{X}$ via

$$\tilde{X}(i, j) = \text{clip}(X(i, j) + \delta, \, 0, 255),$$

where $\delta \in \mathbb{R}^3$ is a fixed color–translation vector. We then run the same style–transfer procedure as in Section 4.4 using both $L$=100 (Figure 7) and $L$=10 (Figure 8) projection directions.

Across both projection budgets, methods that employ randomized QSW constructions or BO–based refinement accurately recover the color shift and yield very small final $W_2$ distances, while deterministic QSW variants show larger variability. As in the main experiments, **ARBOSW** behaves

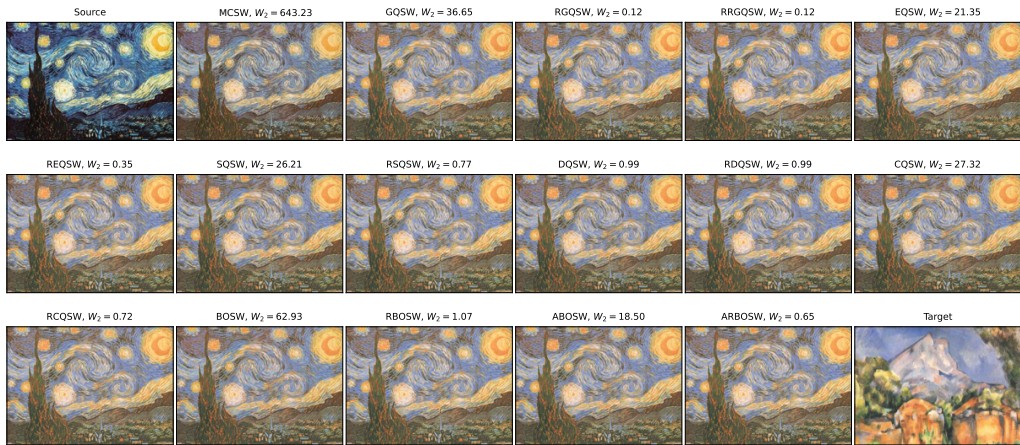

Figure 5: Full comparison for image style transfer with $L$=100.

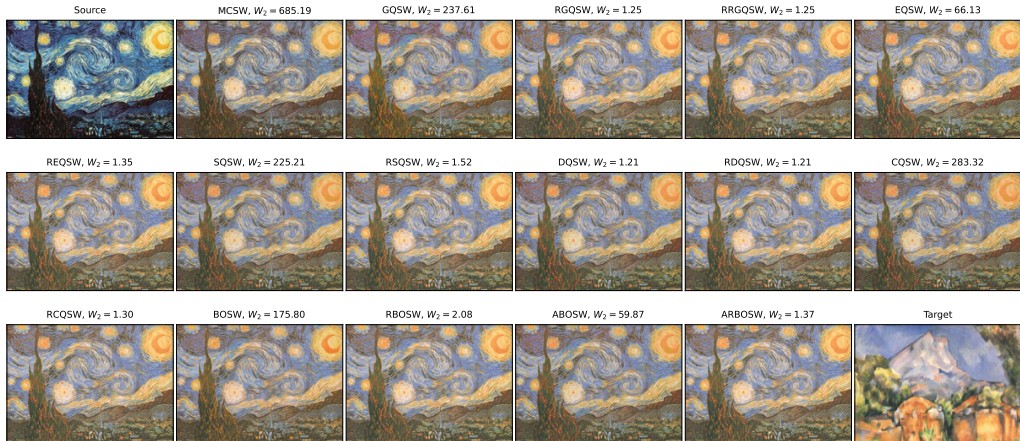

Figure 6: Full comparison for image style transfer with $L$=10.

comparably to the strongest randomized baselines at $L$=100, and **RBOSW** improves over vanilla SW though it is typically outperformed by RCQSW and RGQSW. These controlled translation results further highlight the stability of the BO–refined direction sets and their consistency across varying projection budgets.

### A.6    FULL COMPARISON FOR DEEP POINT-CLOUD AUTOENCODERS

**Setting.**    We follow the same setup as in Section 4.5: ShapeNet Core-55 training, PointNet encoder/decoder, $L = 100$ projection directions, SGD with learning rate $10^{-3}$, momentum 0.9, weight decay $5 \times 10^{-4}$, batch size 128, and training for 400 epochs. Evaluation is performed on Model-Net40, reporting $SW_2$ and $W_2$ reconstruction losses averaged over three runs. Here, we expand the comparison to include the entire QSW family (EQSW, GQSW, SQSW, DQSW, CQSW) and their randomized counterparts (RQSW, RRGQSW, REQSW, RREQSW, RSQSW, RDQSW, RCQSW), alongside our BO-based methods (BOSW, RBOSW, ABOSW, ARBOSW).

**Results.**    Figure 9 shows qualitative reconstructions across all baselines and our BO variants. While randomized QSW methods produce consistent and visually plausible outputs, our BO-based designs, especially **ABOSW**, stand out with sharper reconstructions and more stable geometry. Notably, CQSW remains a strong deterministic baseline, but **ABOSW surpasses it** in final reconstruc-

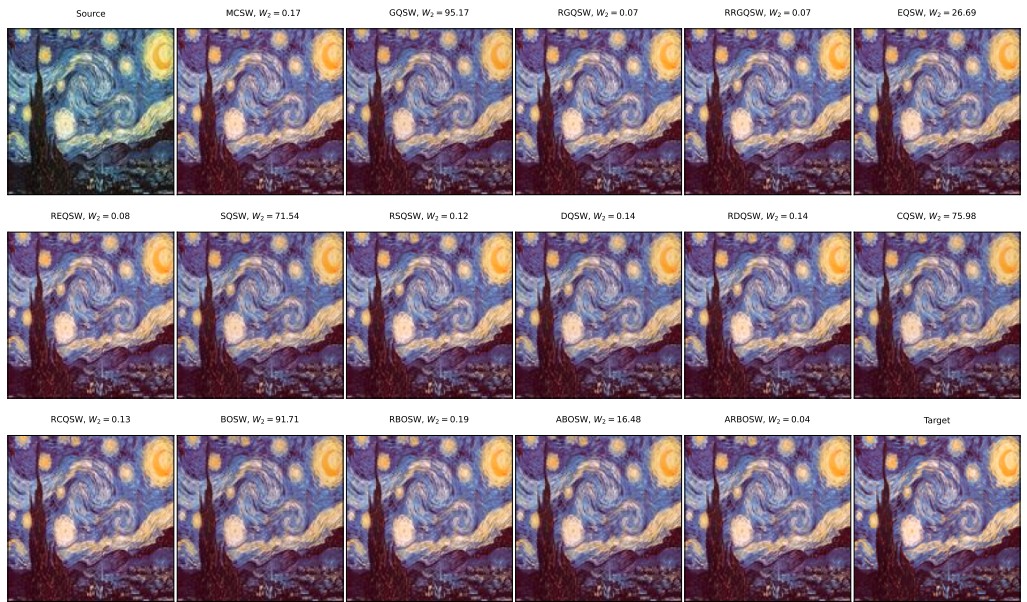

Figure 7: Color–translation experiment at $128 \times 128$ with $L = 100$.

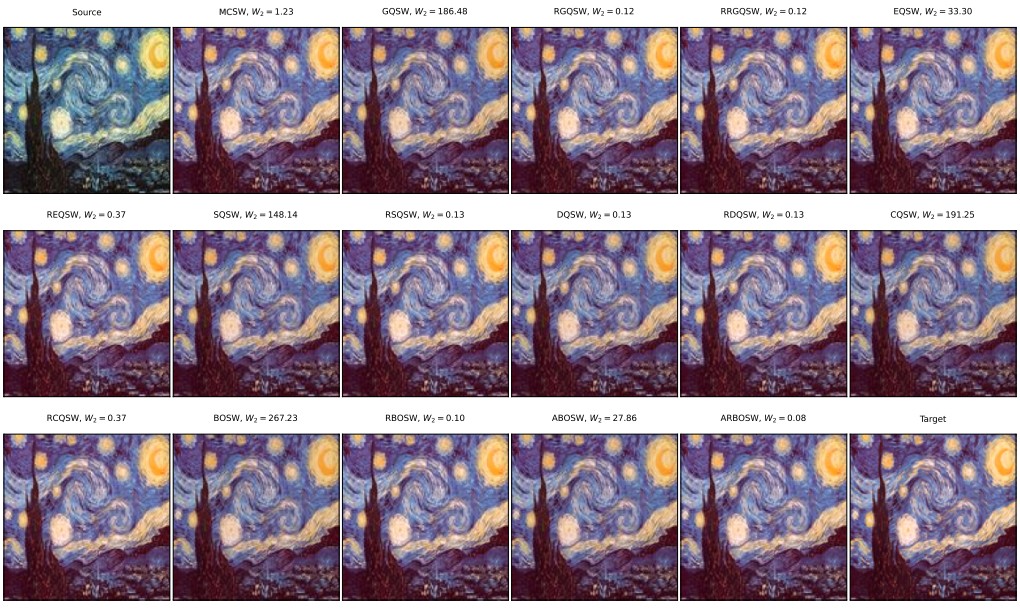

Figure 8: Color–translation experiment at $128 \times 128$ with $L = 10$.

tion loss while maintaining visual quality. This contrasts with gradient flow experiments, where refresh-style variants (RBOSW/ARBOSW) had stronger performance; in autoencoders, the stable large-scale dataset favors consistent one-shot or hybrid projection sets (BOSW/ABOSW). These observations reinforce our conclusion that the optimal projection strategy is task-dependent: stability-driven tasks benefit from non-refresh BO, while dynamic flows gain from refresh hybrids.

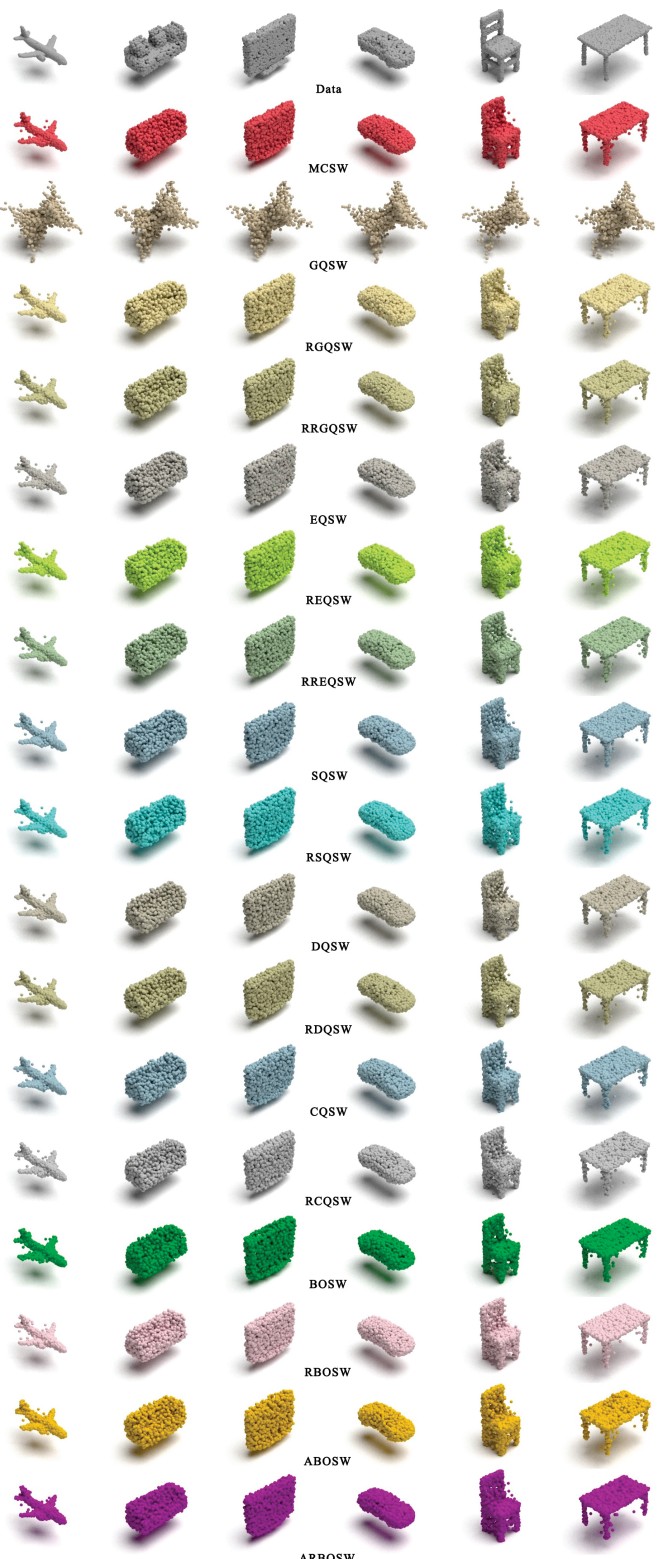

Figure 9: Full reconstructed point-clouds from MCSW, QSW, RQSW, and BO variants with L = 100.

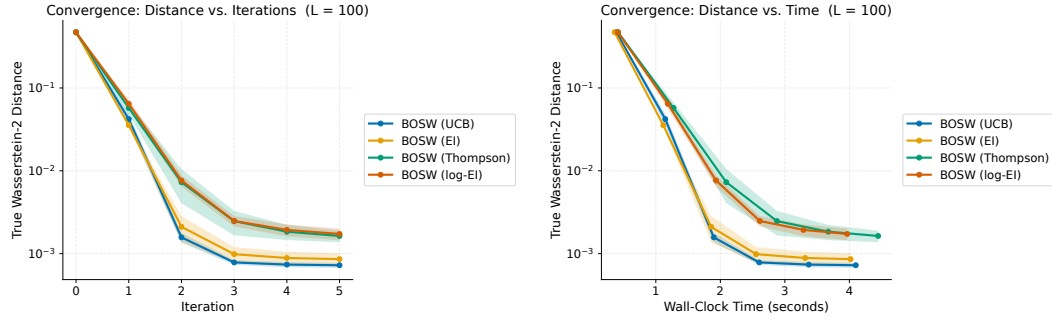

(a) Convergence with increasing iterations.

(b) Convergence with increasing wall-clock time.

Figure 10: Ablation study of different acquisition functions for Bayesian optimization, including upper confidence bound (UCB), expected improvement (EI), Thompson sampling, and log-EI. UCB consistently performs best in terms of number of iterations and wall-clock time required to converge.

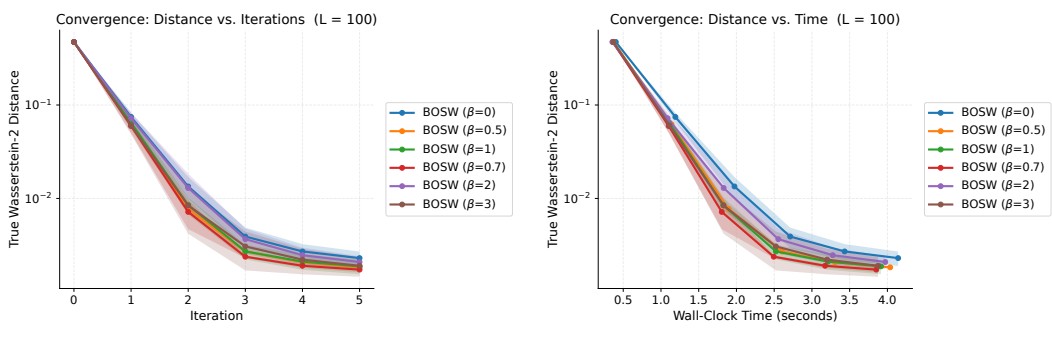

(a) Convergence with increasing iterations.

(b) Convergence with increasing wall-clock time.

Figure 11: Ablation study of different values of $\beta$ (the weight parameter for the UCB acquisition function). Although the different values of $\beta$ yield similar results, $\beta = 0.7$ consistently demonstrates the best performance.

## B  COMPUTATIONAL INFRASTRUCTURE

We use dual NVIDIA RTX A6000 GPUs to conduct experiments on training deep point-cloud autoencoders. Other applications are run on a MacBook Pro 14-inch (Nov 2023) equipped with an Apple M3 Max chip and 36 GB memory.

## C  ABLATION STUDIES

Section 3.1 describes several modeling and parameter choices made for our implementation of Bayesian optimization. We justify these choices here via ablation studies. Each study is evaluated on the point-cloud interpolation (gradient flow) example of Section 4.3. For consistency, we also select $L = 100$ for all our ablation studies unless otherwise noted.

First, we experiment with different possible acquisition functions for Bayesian optimization. We test popular functions such as upper confidence bound (UCB), expected improvement (EI), Thompson sampling (Thompson, 1933), as well as the recently-proposed log-EI (Ament et al., 2023). Figure 10 shows that UCB consistently outperforms other acquisition functions. Thus, we select UCB for all of our examples in the paper. Interestingly, we note that a recent work (Vardhan et al., 2024) also found that LCB (identical to UCB, but for minimization rather than maximization) was the most efficient acquisition function choice, in a completely different application domain (design optimization).

Next, we consider different values of $\beta$, the parameter that tunes exploration vs. exploitation in the UCB acquisition function. (Note that in our experiments, we use a constant $\beta$; adaptive $\beta$

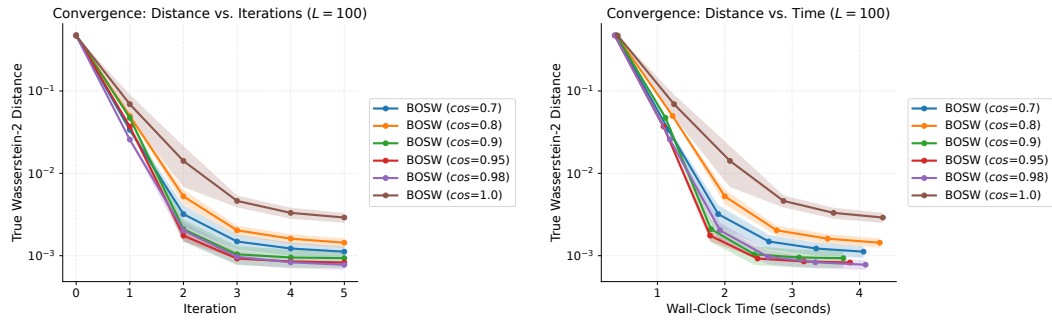

(a) Convergence with increasing iterations.   (b) Convergence with increasing wall-clock time.

Figure 12: Ablation study of different values of the cosine-similarity cutoff threshold used in our method. The plots motivate our usage of $0.98$ for all our experiments.

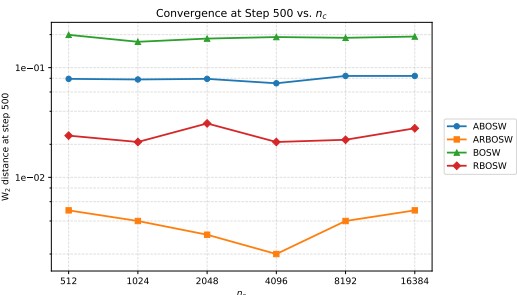

Figure 13: Effect of varying $n_c$ on the convergence of our BO-based methods. For most methods, $n_c = 4096$ is optimal, except for BOSW, which attains minimal error after 500 steps using $n_c = 1024$. Note that the vertical axis is scaled logarithmically to make the data points easier to distinguish.

schedules are certainly possible and may benefit bias (see Appendix D.) As demonstrated in Figure 11, the constant value $\beta = 0.7$ that we select for all our experiments outperforms all other choices considered (both larger and smaller).

We also explore different thresholds for the cosine-similarity cutoff (used to reject the selection of directions that are too close to each other). We remark that this value should typically be chosen closer to 1 as $L$ increases, to facilitate large numbers of slices becoming dense on the unit hypersphere. For $L = 100$, though (as generally used in our experiments), Figure 12 shows that our choice of $0.98$ is near-optimal; while sometimes a slightly lower value ($0.9$ or $0.95$) helps achieve a slightly lower error at a particular time, a value of $0.98$ results in the lowest error after sufficient iterations or time. An annealing scheme for a dynamic threshold value may unlock marginally better performance (particularly for large $L$), but we feel this would be a very minor optimization, especially since $L$ is usually kept relatively small to maintain computational efficiency of SW.

Next, we consider different values of $n_c$, the number of candidate directions considered on any iteration of our core BO routine. We chose $n_c = 4096$ to copy the number of candidate directions used in the QSW methods of Nguyen et al. (2024a). Interestingly, as Figure 13 shows, this value turns out to be optimal for all methods except BOSW (where 1024 is optimal). Nonetheless, we found that values from 512 to 16384 did not significantly change the error in our methods after 500 steps, despite the linear-time computational complexity associated with $n_c$. Thus, in practice, it may be appropriate to use smaller values of $n_c$, accepting a modest change in accuracy for a potentially substantial change in performance. We note that, as expected, for sufficiently small values of $n_c$, no method converged well (neither QSW methods nor our BO-based methods).

Finally, we consider different values of $R$ for the variants of our method that use periodic BO refreshes. We evaluate $R \in \{5, 10, 25, 50\}$ for both RBOSW and ARBOSW. Smaller $R$ values adapt more aggressively but incur substantial GP and BO overhead, while larger values refresh less

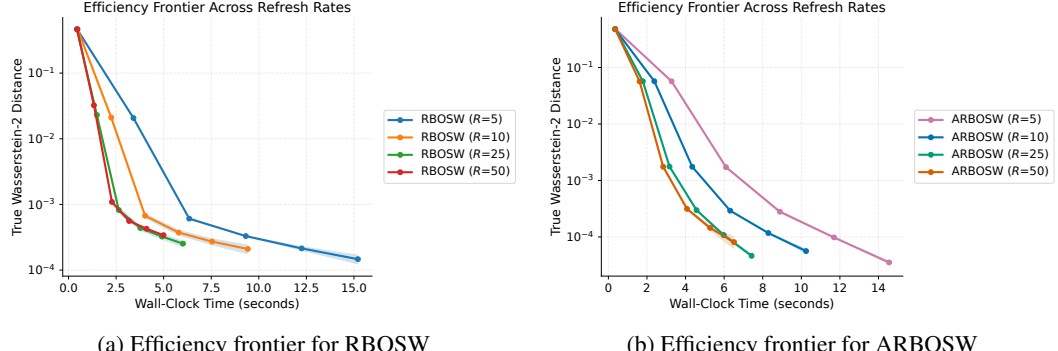

(a) Efficiency frontier for RBOSW  (b) Efficiency frontier for ARBOSW

Figure 14: Ablation study on refresh frequency $R$ for BO-refined SW estimators. Across both RBOSW and ARBOSW, refresh rates in the range $R = \{5, 10, 25, 50\}$ provide the best accuracy–efficiency balance, with $R = 25$ achieving the strongest performance overall.

often and risk using stale direction sets. As illustrated in 14, $R = 25$ consistently offers the best accuracy–efficiency trade-off: it converges quickly while maintaining low overhead. A slightly less frequent refresh, $R = 50$, performs comparably and is a reasonable secondary choice when computation time is constrained. In contrast, the more frequent refresh schedules ($R = 5$ and $R = 10$) incur significant overhead and have the poorest wall-clock efficiency. Motivated by these results, we adopt $R = 25$ for all of our numerical experiments.

## D  THEORETICAL PERFORMANCE OF BAYESIAN OPTIMIZATION FOR SLICED WASSERSTEIN

Although our paper is primarily intended to provide numerical evidence of the efficacy of using Bayesian optimization for sliced Wasserstein distance computations, we briefly remark on the possibility of theoretical performance bounds.

It is known that under i.i.d. MC sampling and sufficient regularity assumptions, SW converges at a rate of $O(L^{-1/2})$. By the triangle inequality, we can relate the error of our BO methods to the error of MCSW:

$$\left| SW_L^p(\hat{\Theta}_T) - SW_p^p(\mu, \nu) \right| \leq \underbrace{\left| SW_L^p(\hat{\Theta}_T) - SW_L^p(\Theta_{\mathrm{MC}}) \right|}_{\text{BO-MC Error}} + \underbrace{\left| SW_L^p(\Theta_{\mathrm{MC}}) - SW_p^p(\mu, \nu) \right|}_{\text{MC-SW Error}}.$$

The latter term is bounded above by $O(L^{-1/2})$, while bounding the former term remains an open question. The primary reason bounding that term is problematic is that BOSW may *not* perform as well as MC SW in the limit, because BOSW is not generally an unbiased estimator. For instance, with our UCB acquisition function as used in our paper,

$$\alpha_t(\theta) = \mu_{t-1}(\theta) + \beta \sigma_{t-1}(\theta),$$

we take a constant $\beta = 0.7$. Thus, $\alpha_t$ will always be encouraged to select $\theta$ yielding high $f(\theta; \mu, \nu)$. In the limit, this suggests that

$$\frac{1}{L} \sum_{i=1}^{L} f(\hat{\theta}_i) \to \sup_{\theta \in S^{d-1}} f(\theta) > \mathbb{E}_{\theta \sim U}[f(\theta)].$$

This is a strict inequality assuming $f$ is non-constant on $S^{d-1}$; and when sampling $\theta$ uniformly (MC), as mentioned above, $\mathbb{E}_{\theta \sim U}[f(\theta)] = 0$. This rough sketch demonstrates that BOSW, at least with UCB with a constant $\beta$, will not converge in the limit—despite its attractive numerical performance (demonstrated in our experiments in the main body) during early iterations. Bounding the performance of BOSW for finite $L$ is an interesting open question. It may be possible to approach this question using the concepts of information gain and regret minimization; see, e.g., Vakili et al. (2021). This would be impacted by our choice of using an angular RBF kernel.

Despite gaps in the current theory, we note that we can use an annealing schedule for $\beta$ to drive our GP-UCB sampler towards eventual uniform coverage of the sphere. In spirit, this follows the idea of using BOSW for early iterations, then switching to MC or other variants for later iterations—an idea that could harmonize the benefits of fast early convergence with BO-based methods with the benefits of uniform coverage of MC samplers. For instance, if we define

$$\hat{\alpha}_t(\theta) = \epsilon_t \alpha_t(\theta) + (1 - \epsilon_t)\frac{1}{|S^{d-1}|},$$

we can choose $\epsilon_t = t^{-\gamma}$ ($0 < \gamma < 1$). Then as $t \to \infty$, $\epsilon_t \to 0$, but $\sum_t \epsilon_t = \infty$, so informally speaking[2], the second Borel–Cantelli lemma would suggest that every measurable region of the hypersphere is sampled infinitely often with probability 1. It follows that the bias will match that of uniform MC sampling, which is 0 in the limit; and under this scheme, the convergence rate would reach $O(L^{-1/2})$ asymptotically (whether it is faster or slower during initial iterations).

## E   USE OF LARGE LANGUAGE MODELS

Large language models (LLMs) were used in the preparation of this paper. Specifically, Perplexity was prompted to act as a peer reviewer for our draft. In this context, the model identified several recent references for us to discuss in our related work that we previously had not included; and the model reminded us to perform a few of the ablation studies seen in Appendix C.

---

[2]The lemma assumes each "event" (direction selection) is independent, which is not the case, by design. Nonetheless, this assumption *would* hold in the limit of $\epsilon_t \to 0$. Generalizations of the Borel-Cantelli lemma that do not require strict independence have been studied (Chandra, 2008; Biró & Curbelo, 2021), and it is possible that those may apply here. For the present work, though, we only characterize our arguments as informal.

