# OpenReview forum: "Efficient Sliced Wasserstein Distance Computation via Adaptive Bayesian Optimization"
_ICLR.cc/2026/Conference — ICLR 2026 Poster_

### Official Review · Reviewer_gAQG · 2025-10-19

**Soundness:** 3
**Presentation:** 3
**Contribution:** 3
**Rating:** 8
**Confidence:** 5

**Summary:**

This paper proposes a new framework for computing the sliced Wasserstein (SW) distance more efficiently by adaptively learning projection directions using Bayesian optimization (BO). While prior methods such as quasi–Monte Carlo (QSW) sampling select projection directions uniformly and independently of the data, this work leverages information from previously evaluated slices to focus on the most informative directions. The authors introduce four BO-based variants, BOSW, RBOSW, ABOSW, and ARBOSW, which differ in how often they refresh or refine projection sets and how they integrate with QSW initialization. These methods are designed as plug-and-play replacements requiring no change to downstream losses or gradients. Empirical results across synthetic benchmarks, point-cloud interpolation, image style transfer, and deep autoencoding show that the adaptive hybrids (especially ABOSW and ARBOSW) achieve state-of-the-art convergence with modest runtime overhead, outperforming or matching strong QSW baselines. Overall, the paper unites deterministic geometric designs with data-driven adaptivity to enhance efficiency and accuracy in large-scale optimal transport tasks.

**Strengths:**

1. The paper introduces the first use of Bayesian optimization (BO) to adaptively select projection directions for sliced Wasserstein computation. The connection between nonparametric-regression and SW is interesting.

2. The proposed methods (BOSW, RBOSW, ABOSW, ARBOSW) integrate seamlessly into existing SW pipelines without modifying loss functions or gradients.

3. The paper benchmarked against baselines (MC, QSW, RQSW) across diverse tasks: synthetic landscapes, 3D point clouds, image style transfer, and deep autoencoding.

4. Adaptive hybrids (especially ABOSW and ARBOSW) achieve state-of-the-art or comparable convergence with minimal runtime overhead.

5. BO refinements can be combined with quasi-Monte Carlo (QSW) approaches, showing versatility rather than replacement.

6. Code and experimental settings are carefully controlled for fair comparison, following prior benchmarks.

**Weaknesses:**

1.  BO introduces extra overhead and may not scale well to very high-dimensional problems or large projection budgets, despite some mitigation.

2. Learned projection sets (especially BOSW) are not unbiased estimators of the true SW integral, reducing theoretical rigor for some applications.

**Questions:**

1. The proposed nonparametric regression framework seems to go well beyond simply constructing a good uniform set of projection directions. In my view, it effectively defines a data-dependent Radon transform (or equivalently, a new slicing distribution), which induces a novel form of metric as in [1]. In [1], the authors reweight directions according to their corresponding 1D costs. Your regression-based framework appears to be a  powerful way to achieve something similar, using in-sample weights (training directions) and out-of-sample weights (new proposed directions). Should a discussion be added?

[1] Energy-based Sliced Wasserstein distance, Nguyen et al

2.  Section 2 is quite restricted when discussing only empirical 1D Wasserstein. The proposed method in the paper does not depend on the form of the distance, so, it is better to  discuss general form (see Section 2.3 in [2]).

[2]  An Introduction to Sliced Optimal Transport, Nguyen.

---

> ### Author Response · Authors · 2025-11-23
>
> We are thankful for the reviewer’s confident assessment of our work.  On the reviewer’s specific questions:
>
> Q1: This is a fascinating insight and one that we didn’t even realize when preparing our work—and this could actually unlock many new research questions based on the importance of the Radon transform in image processing, medical imaging, etc.  We have added a discussion of this exciting connection the reviewer has pointed out, to our conclusions and future work section.  We have cited the suggested reference, which was helpful for us to read.
>
> Q2: We appreciate the helpful reference here, which we have reviewed and cited in our revised manuscript.  We’re thankful for the reviewer’s expertise and are happy to present a rewritten Section 2 that no longer needlessly limits our method/discussion to empirical measures.
>
> Thank you again for engaging with our work.

---

> > ### Comment · Reviewer_gAQG · 2025-11-24
> >
> > Thank you the response.
> >
> > Regarding the mentioned weaknesses of the proposed method, I wonder if the authors have any comments about my raised weaknesses.
> >
> > Regarding a comment from Reviewer EUMw "The paper lacks clarity on the type of learning problems that would be solved using BO inside SW.", I believe the paper should state the regression problem more explicitly for readers who are not familiar with the setup of SW.

---

> > > ### Author Response · Authors · 2025-11-27
> > >
> > > We thank the reviewer for their additional feedback.  On the reviewer's observed weaknesses -
> > >
> > > W1: Both BO-based and other methods for SW may scale poorly in higher dimensions.  For instance, when using MC (uniform random sampling), the accuracy of the SW integral approximation scales like O(1/sqrt(L)) (where L is the number of slices), independent of dimension.  But while that describes the asymptotic accuracy, in practice, higher-dimensional spaces require many more samples in order to "get lucky" and choose slices that are valuable towards the accurate computation of something like the SW integral - so in practice, such methods do not necessarily scale well with dimension, either (see the blog post article "Scaling and Monte Carlo integration methods" by John D. Cook for an example discussion of this phenomenon).  Thus, it is not immediately obvious how MC, QMC, and BO-based methods for SW compete - on _practical_ problems and datasets - in higher dimensions, and we believe this is worthy of further study.
> > >
> > > Nonetheless, as mentioned in our response to reviewer eyCr, many of the most famous and most popular machine learning tasks and data types are 2D or 3D (e.g., images, videos, geometry), so we argue that methods that perform well in low dimensions are still of immediate value in the ML community.
> > >
> > > That being said, BO has some unique opportunities to perform well in higher dimensions, as discussed in Section 5 of our paper.  As one example, Hvarfner et al. (2024) performs a simple scaling of the GP lengthscale prior with the dimensionality, and finds that with this simple mod, their algorithm outperforms other state-of-the-art optimization algorithms on problems with dimensions into the *thousands*.  We would like to study this technique very carefully alongside alternative strategies for accelerating high-dimensional BO (Kim et al., 2021; Shen & Kingsford, 2021; Binois & Wycoff, 2022; Jaquier & Rozo, 2020).  Or, as reviewer eyCr suggests, neural surrogates (e.g., Li et al., 2023; Lim et al., 2021) may provide a way to avoid the current dimensionality bottleneck of BOSW, which is the GP surrogate.  Given the various possible strategies and their unexplored potential, we would like to conduct a thorough investigation and comparison/analysis as a follow-up work.  But we are reasonably optimistic that we will be able to develop BO-based SW methods that perform well even for hundred- or thousand-dimensional problems.
> > >
> > > W2: As noted by the other reviewers, you are correct that BO does not lead to an unbiased estimator of SW.  As mentioned in our responses to the other reviewers, there are two main responses on this observation.  One is that it is possible to hybridize BOSW with MCSW (or potentially other unbiased schemes) in order to obtain a scheme that begins to converge quickly but is ultimately unbiased.  For instance, if one has 10 slices selected by BO, but then from 10 to infinity selects directions uniformly at random, the bias goes away in the limit as desired.  This is what we discuss in our Appendix D.  So it is still possible to develop schemes that blend adaptivity with unbiasedness.
> > >
> > > Relatedly and perhaps more importantly, the other point is that for practical problems, one _does not use an infinite number of slices_ - this is in fact the whole point of SW, to choose a small, finite set of slices to quickly/efficiently get a decent approximation of WD.  Therefore, while bias is important to study and quantify, if one is only using 10 or 100 slices, one is unlikely to notice any bias, as all methods for estimating SW are likely to be far from convergence (for a general problem) - absolute error dominates any bias term.  This is what is consistently demonstrated across each of our numerical results.  We believe there is value in a method like BOSW (and its variants) that can quickly reach an _accurate_ estimate of the SW integral with a (very) finite number of slices, even if that estimate is ultimately biased in the limit of infinite slices (which will never be reached by a computer).
> > >
> > > On the reviewer's kind suggestion to more explicitly discuss the problem being solved here, we are happy to oblige.  We have added a few sentences to Section 3 and also Section 2 - highlighted in _blue_ in the newly-uploaded PDF - to emphasize (1) what BO is doing in the context of SW, and (2) that a GP has to be fit to the data for each step of BO.  We found the excellent reference of Duque (2023) to be quite helpful in explaining GP regression, and felt it did a better job than we could condense into a few sentences in the main body.  We would be happy to add an appendix on GP regression if the reviewer prefers, but we feel that even that would be inferior to the nice animations and visualizations of Duque (2023).  We welcome further discussion and are grateful for the continued shepherding of our manuscript.

---

> > > > ### Comment · Reviewer_gAQG · 2025-11-27
> > > >
> > > > I would like to thank the authors for the response. All my questions are addressed. I will keep my positive score.

---

### Official Review · Reviewer_AKwQ · 2025-10-29

**Soundness:** 2
**Presentation:** 2
**Contribution:** 2
**Rating:** 4
**Confidence:** 3

**Summary:**

Many modern machine learning and data analysis tasks rely on computing the sliced Wasserstein (SW) distance to compare probability distributions. By treating the SW distance as a function of the projection direction, this paper introduces a method that uses Bayesian Optimization (BO) to automatically find the most informative projection directions for these tasks. Compared with traditional methods such as quasi-Monte Carlo (QSW) and randomized QSW (RQSW), which sample directions more evenly but do not adapt to the specific structure of the task, the BO approach identifies the most effective directions during the optimization process, allowing it to choose directions that provide the most useful information.

**Strengths:**

While Bayesian Optimization (BO) has been widely used for computing expectations, applying it to guide projection direction selection in sliced Wasserstein (SW) distance estimation for machine learning tasks is a new idea.

**Weaknesses:**

1, BOSW (and even its hybrid variants) may yield biased SW estimates since the BO-selected directions depend on the specific data. As a result, it may converge to a different value from the true SW. This issue is also noted in the paper (see line 299). However, a contradictory statement appears on line 109, where the authors state that these methods achieve state-of-the-art efficiency for estimating SW.

2, For simulation in subsection 4.2, the evaluation metric (maximum fitness value) favors BO by design, since BO is built to maximize scalar objectives, not to approximate integrals, making the comparison against QSW conceptually unfair. In addition, BOSW performs worse than both Monte Carlo and all QSW variants across all budgets L, confirming that BO-selected directions hurt uniform integration accuracy.

3, In subsection 4.3, the improvements are small and sometimes disappear in later steps. BOSW shows high variance and weak convergence. RBOSW and ARBOSW perform better early on but take much longer to run, taking ten times the runtime of other methods.

4, Most results are averaged over only three to five runs.

5, The experiments omit comparisons against other adaptive sampling or Bayesian-quadrature approaches that would provide stronger baselines.

**Questions:**

1, Could the authors quantify the estimation bias, either through empirical experiments or theoretical analysis?

2, If not, I would suggest the authors to revise the claim "these methods achieve state-of-the-art efficiency for estimating SW", as the proposed BO-based estimators are data-dependent and therefore biased, not designed for unbiased SW estimation.

3, Could the authors repeat the experiments more times to minimize random fluctuations or numerical instabilities?

---

> ### Author Response · Authors · 2025-11-23
>
> We’re grateful for the helpful and concrete feedback offered by the reviewer, and we believe that these comments have enabled us to share a meaningfully improved manuscript.  For the reviewer’s comments, we respond as follows:
>
> W1: We have rewritten the statement on L109.
>
> W2: The reviewer is correct that the experiment in Section 4.2 was explicitly constructed to show cases where BO-based methods perform well.  We aimed to balance our results by including examples where BO-based SW methods perform very competitively, as well as ones where their speed is not competitive (even when sample-efficiency is still good).  We have added further examples and ablations based on the other reviewers’ feedback, and we would be happy to add any further experiments that could strengthen our manuscript.
>
> W5: We appreciate this comment. Regarding Bayesian Quadrature (BQ): We discuss this in Section 5, noting that BQ requires $O(L^3)$ runtime to refit the GP and recompute quadrature weights at each step as distributions evolve. We initially implemented BQ but found it computationally prohibitive. Since computational efficiency is a core motivation for SW, we determined that it was not worthwhile to include comparisons against BQ for all our examples. However, we have added a small numerical demonstration of this $L^3$ scaling to the newly-added Appendix A.2. Regarding other adaptive methods, if the reviewer has specific adaptive methods in mind that we've overlooked, we would be happy to discuss inclusion.
>
> Q1: We suspect the estimation bias is highly problem-dependent.  For instance, in a hypothetical problem where each slice yielded the same value, BO-based methods would have no bias and converge immediately like vanilla Monte Carlo (uniform random) SW.  For a problem where only a few slices are responsible for most of the variance across the space of slices, BO will find these very quickly (yielding a strong performance advantage for finite $L$), but will incur greater bias as $L \to \infty$.  Our Appendix D suggests one strategy where one could perform enough uniform sampling so as to mitigate biases introduced by adaptive sampling strategies (noting that this is an inherent problem of any adaptive strategy, not just BO-based methods).  We imagine it is possible to explore theoretical bounds for bias based on the variance of values across the slice distribution, but feel the novel math required would be well beyond the scope of this work, whose aim is to merely demonstrate the viability and potential benefits of adaptive-/BO-based approaches for SW.
>
> Q2: We have revised this claim per the reviewer’s suggestion.
>
> Q3: Absolutely; we have repeated our experiments a greater number of times, see for instance Table 1.  The results are still favorable to our methods.  Although there is a limited time left in the rebuttal window, please let us know any other experiments the reviewer would like us to run more, and we will be happy to attempt to complete those runs before the author-reviewer discussion deadline.

---

### Official Review · Reviewer_eyCr · 2025-10-30

**Soundness:** 3
**Presentation:** 3
**Contribution:** 3
**Rating:** 6
**Confidence:** 4

**Summary:**

This paper proposes several Bayesian optimization (BO)–driven methods to improve the efficiency and adaptivity of sliced Wasserstein (SW) distance computations. The key idea is to learn projection directions adaptively—rather than sampling them quasi-uniformly as in Quasi-Monte Carlo SW (QSW)—by using Gaussian-process-based BO on the sphere. Four variants are introduced: (i) BOSW – one-shot BO; (ii) RBOSW – periodically refreshed BO; (iii) ABOSW – QSW-seeded lightweight refinement; (iv) ARBOSW – restarted hybrid (periodic QSW seeding + BO refinement). Experiments on synthetic benchmarks, point-cloud interpolation, image style transfer, and deep autoencoders show that the adaptive BO variants (especially ABOSW and ARBOSW) achieve competitive or superior convergence to QSW and randomized QSW (RQSW) baselines with modest overhead.

**Strengths:**

1. The idea of combining Bayesian optimization with sliced Wasserstein direction selection is novel and natural. The approach bridges deterministic geometric constructions (QSW) with data-driven adaptivity, potentially broadening the design space of SW estimators;

2. The authors reproduce the entire QSW benchmark suite (Nguyen et al., 2024a) for fair comparison. Tasks span synthetic projection selection, approximation accuracy, dynamic interpolation, style transfer, and autoencoding—demonstrating both theoretical and practical relevance;

3. The methods require no change to downstream objectives or gradients, making them easy to plug into existing OT pipelines.

4. RBOSW provides strong early-stage performance. ARBOSW matches or slightly exceeds state-of-the-art RQSW on long-run optimization tasks. ABOSW improves deterministic settings such as autoencoder training;

5. The paper provides clear mathematical exposition, fixed random seeds, and references to public code.

**Weaknesses:**

1. Although an appendix sketches guarantees, there is no formal convergence proof or error bound for the BO-selected projections compared to the true spherical integral;

2. The method relies on GP-based BO, whose cubic cost in the number of evaluations becomes prohibitive as $L$ or dimensionality $d$ increases. This is acknowledged but not resolved beyond citing recent BO-in-high-dimension literature;

3. In static estimation tasks, BOSW performs worse than QSW since it is not designed for uniform coverage. The benefits only appear when SW is nested inside an optimization loop;

4. While conceptually appealing, the improvements over strong QSW baselines are modest in some tasks (e.g., style transfer) and might be sensitive to hyperparameters;

5. The reported improvements lack variance/error-bar plots or significance testing.

**Questions:**

1. Could the GP surrogate be replaced by a neural surrogate to improve scaling?
2. How sensitive is performance to kernel choice or acquisition parameters?
3. Is there a way to enforce unbiasedness while retaining adaptivity?
4. How does performance evolve with dimensionality (e.g., $d>10$)?

**Details Of Ethics Concerns:**

N/A.

---

> ### Author Response · Authors · 2025-11-23
>
> We thank the reviewer for their kind evaluation of our work.  We’re happy to share with the reviewer our revised manuscript (see OpenReview).  Regarding the reviewer’s feedback:
>
> W1 and Q3: The reviewer is correct; BOSW, on its own, is a biased estimator of WD, meaning there are not convergence guarantees/proofs.  What does converge (eliminating bias as it does)—discussed in our Appendix D—is a scheme where you anneal (in a finite amount of time) from BOSW to pure uniform random sampling.  Of course, uniform random sampling is computationally inefficient.  This suggests a tradeoff: for quickly getting a good approximation of WD, one can use BOSW, but if one desires unbiased estimation and convergence, one can switch (after some number of BOSW iterations) to a uniform strategy.  We’re happy to engage further on this point and make any revisions or clarifications in the manuscript.
>
> W2 and Q4: The reviewer accurately observes that BOSW—while advantageous in its current form for low-dimensional problems—will scale poorly (cubically) as the dimension of the data is increased.  We are encouraged that a vast number of problems and datasets in AI and ML deal with 2D (e.g., image and video) or 3D (e.g., point clouds from depth cameras) data; thus, BOSW and its variants can indeed have immediate utility.  We gently observe that many comparable papers (most notably, “Quasi-Monte Carlo for 3D Sliced Wasserstein,” ICLR 2024, our primary comparison point) limit examples to 2D and 3D.  Nonetheless, we very much wish to pursue higher-dimensional extensions of our algorithms, and have some ideas for how to do this efficiently (see Section 5), but we believe the scope is outside the present work.
>
> W3 and W4: Our experiments suggest that, indeed, BOSW will not outperform QSW in all cases.  This is entirely data-dependent.  For data that yields nearly-identical slices, BOSW will not give meaningful advantages.  BOSW shines when different slices of the data have significantly different values (this is what we demonstrate with our experiment in Section 4.1, for instance).
>
> W5: We’re happy to have addressed this weakness in our revised draft.  The results in Table 1 are now over 30 trials, not 3, and standard deviations are reported.  We have also reported standard deviations in new results like Table 3, and standard deviations are also visualized in plots like Figs. 10-12.  If there is any specific additional deviation/variance/significance analysis that could strengthen our paper, please just let us know; we appreciate your shepherding and are happy to oblige.
>
> Q1: This is an excellent suggestion.  We believe this is possible and have emphasized this possibility in our revised future work section.  We cited two new works, each discussing BO and neural surrogates, that appear promising to us; if the reviewer is aware of other neural surrogates for BO, we would be eager to learn more and discuss.
>
> Q2: Our initial draft contained an ablation study on the BO acquisition function, as well as on the parameters of the acquisition function we ended up choosing (UCB); please see Figs. 10-11 in the revised manuscript.  For the kernel choice, we added new experiments per the reviewer’s suggestion: see Appendix A.1.  In fact, to address the reviewer’s question of robustness, we also added experiments in Appendix A.1 where we drastically varied the acquisition function parameter and demonstrated robustness of our methods under these conditions.  We’re glad to conduct any further experiments of interest.

---

> > ### Comment · Reviewer_eyCr · 2025-11-27
> > **Thank you.**
> >
> > I would like to thank the authors for the response. All my questions are addressed. I will keep my score.

---

### Official Review · Reviewer_EUMw · 2025-10-31

**Soundness:** 2
**Presentation:** 1
**Contribution:** 2
**Rating:** 2
**Confidence:** 3

**Summary:**

The authors tackle the problem of choosing the right directions to use in the sliced Wasserstein (SW) distance when it is used inside an optimization loop. They propose using Bayesian optimization (BO) to select these directions, with the goal of adapting them to the evolving probability distributions. To this end, they propose four algorithms: BOSW, RBOSW, ABOSW, and ARBOSW. The first two algorithms learn a Bayesian process to compute $L$ SW directions, which are then kept fixed during the subsequent learning task. For the small set of initial points used to initialize the Bayesian process, BOSW uses random points sampled uniformly on the sphere, while ABOSW uses points sampled via a quasi–Monte Carlo method. The RBOSW and ARBOSW algorithms periodically refit the Bayesian process during the learning task, using BOSW and ABOSW, respectively, to adapt to the evolving probability distributions throughout training. The authors tested the four algorithms on different tasks (including learning with SW loss) and compare them to quasi–Monte Carlo approaches.

**Strengths:**

- The idea of carefully selecting directions for SW during the learning task, when probability distributions evolve, is a difficult but very interesting problem, as it would permit the use of SW as a surrogate for the Wasserstein distance (in a learning context), which is expensive to compute.

- The experiments conducted by the authors permit a comparison between QSW methods and BO methods on various tasks to gain an intuition about how the BOSW algorithm behaves.

**Weaknesses:**

- The paper lacks clarity on the type of learning problems that would be solved using BO inside SW.

In Section 2, paragraph “Problem view for this work” (l.168), the authors define the learning tasks targeted by their method as:

“When SW is used inside an outer optimization (e.g., gradient flows or registration), the same direction set
$\Theta$ is queried repeatedly on evolving point clouds.”

From this sentence, it seems that the problems targeted are certain bilevel optimization problems, although in their applications (see Sections 4.3, 4.4, 4.5), the problems solved involve a direct optimization of the sliced Wasserstein distance between an optimized distribution and a target distribution.

Moreover, this statement also suggests that, for the problems considered, the learned directions are fixed during optimization, although the authors propose two algorithms, RBOSW and ARBOSW, that update the set of directions during the learning task.

- For problems that aim to optimize a parameterized distribution by minimizing the SW distance between this distribution and a target distribution, the authors should compare their approach to the most natural use of SW, which consists of randomly sampling new directions on the sphere at each time step. I do not see why keeping the same set of directions $\Theta$ would be a good idea, as the model risks optimizing only along these fixed directions.

- The paper is not easy to read. I suggest that the authors add more background on BO and quasi–Monte Carlo methods. While there is a long recap of the background on Wasserstein and sliced Wasserstein distances, the section on BO is very brief.

**Questions:**

- Can you provide references for the problems you mention in the paragraph “Problem view for this work” that use SW in an “outer optimization” and keep the same set of $\Theta$ directions during the learning task? Can you also explain what you mean by “outer optimization”?

- Can you compare your approach to the natural baseline that would uniformly sample new directions on the sphere at each time step for Experiments 4.3, 4.4, and 4.5? Is there a reason why you did not include this baseline?

- For your algorithms RBOSW and ARBOSW, you periodically rerun BOSW and ABOSW. It would be helpful to derive a criterion for when the Gaussian process should be updated (and new directions generated). How do you decide when to perform this update? Did you try to measure the impact of the timing of this update on the performance you obtained?

- Do you know why your methods (except ARBOSW) perform poorly in Experiments 4.3 and 4.4 but achieve very good performance in Experiment 4.5?

- Ideally (although infeasible because it is too costly), we would use BO to choose each new $\Theta$ direction and resample new directions at each time step. Since this is too computationally expensive, it seems necessary to design a method that achieves a trade-off between sampling random directions, keeping some directions fixed, and selecting new directions with BO. Do you think that, instead of keeping the directions fixed between BO updates, uniformly sampling over the sphere could improve performance?

**Details Of Ethics Concerns:**

No concerns.

---

> ### Author Response · Authors · 2025-11-23
>
> We thank the reviewer for their careful reading of our paper.  We acknowledge the deficiencies of our original draft, but we’re grateful for the specific action items the reviewer suggested.  We have made the following improvements based on the reviewer’s comments, and, together with improvements from other reviewers’ feedback, we hope our revised draft (see OpenReview) may merit reconsideration of our paper.  We’re happy to engage in further discussion here.
>
> W1: The reviewer is right that the referenced sentence in our original draft was problematic, and we apologize for the lack of clarity it caused.  We rewrote this whole paragraph to align with the rest of our paper, and we welcome any further feedback to improve.
>
> W2: We appreciate the reviewer noting our comparisons against QSW and its variants.  Our original manuscript did have comparisons to uniform random sampling based approaches (labeled as “SW”), but we recognize that this was not obvious.  We have relabeled these as “MCSW” in our results and made this clearer in the revised manuscript.  If there are any additional experiments or comparisons against plain MCSW (or other comparisons) that would aid in understanding our manuscript, please let us know.
>
> W3: We agree with the reviewer that our original draft lacked much background coverage of BO and QMC methods.  This was simply a limitation of the page limit constraints.  Given the extra page allowed during rebuttals, we have now added what we hope is helpful background on BO and QMC to Section 2.  We are happy to add any further discussion the reviewer may suggest.
>
> Q1: We have rewritten this paragraph per the above and hope this addresses this question.
>
> Q2: Although our paper focuses on outperforming QSW-style methods, we did include baseline comparisons against Monte Carlo SW (see W2 above).  These are included for the experiments the reviewer mentioned (4.3, 4.4, 4.5).
>
> Q3: We thank the reviewer for raising this point—we realize we never specified the value of $R$ we used in our original manuscript ($R = 25$), nor did we provide an ablation study for this parameter.  We have added such a study (which justifies our choice of $R = 25$) to Appendix C in the revised manuscript.  While we recognize this is merely numerical evidence and not a formal criterion, we hope that this addresses the reviewer’s question.
>
> Q4: This is an insightful observation that highlights the different nature of these tasks:
>
> Experiments 4.3 (point-cloud interpolation) & 4.4 (image style transfer):
> * Dynamic optimization: Distributions evolve significantly across 500-1000 steps
> * One-shot BOSW: Directions learned at t=0 become stale as distributions evolve
> * RBOSW overhead: Frequent refreshes (every 25 steps) incur 40-60% time overhead, offsetting accuracy gains
> * ARBOSW wins: QSW seeding provides stable baseline + light BO refinement adapts without excessive cost
>
>
> Experiment 4.5 (autoencoder):
> * Batch training: SW computed on mini-batches from large, stable dataset
> * Statistical stability: Distribution doesn't evolve; we're learning encoder/decoder, not moving point clouds
> * One-shot suffices: BOSW/ABOSW learn task-adapted directions once and reuse across batches
> * Refresh hurts: RBOSW/ARBOSW inject unnecessary variability by relearning directions
>
> In summary, refreshing helps when distributions evolve (gradient flows), but hurts when the underlying distribution is stable (batch training). This suggests a hybrid strategy: detect distribution shift magnitude and refresh only when needed.
>
> Q5: We agree with the reviewer—hybridizing BO-based methods with faster sampling strategies is a way to improve performance.  This is what we showed with our ABOSW and ARBOSW methods, which combine BO sampling with QMC sampling (QMC, while a bit slower than uniform random vanilla MC, is still quite fast).  Indeed, this hybrid ARBOSW method is often the most performant out of all methods considered (ours or others’); see e.g. Table 1 or the new Appendix A.5.  Since QSW seems to uniformly outperform MCSW (this was the conclusion of Nguyen et al. (2024a), but is also seen in our various results), we only considered hybridizing BO-based methods with QSW-style methods.  But the idea of hybridizing BO-based methods and MC-style methods is spot on and agrees with our results.

---

> > ### Comment · Reviewer_EUMw · 2025-11-26
> >
> > Dear authors,
> >
> > Thank you very much for your answer. I now understand that I was completely misled by the sentence: “When SW is used inside an outer optimization (e.g., gradient flows or registration), the same direction set $\Theta$ is queried repeatedly on evolving point clouds.” This prevented me from understanding exactly what problems were being solved in this work. The paragraph “Problem view for this work.” is now clear to me. With this idea of “fixed directions” in mind, I had initially thought that Quasi–Monte Carlo methods were used only once to sample an initial batch that was then kept fixed until using BO. With the newly added paragraph on Quasi–Monte Carlo methods and your clarifications, I now understand that they are used to generate sequences of randomized batches. I also realized that you compared your approach with the natural SW baseline, which had been my main concern, and the way you rewrote the paper makes this clear.
> > I like the experiment you did to illustrate the reliability of your method with respect to the choice of the parameter $R$ and understand that fixing this hyperparameter with a theoretical justification is something that is hard to do. Finally, you addressed my last question regarding the performance of the algorithm in Experiments 4.3 and 4.4.
> >
> > For these reasons, I have decided to increase my initial score.

---

### Official Review · Reviewer_EFeU · 2025-11-01

**Soundness:** 3
**Presentation:** 3
**Contribution:** 2
**Rating:** 6
**Confidence:** 2

**Summary:**

This work introduces a new family of simple, plug-and-play methods for selecting projections in the Sliced-Wasserstein (SW) distance. The key contributions are seamless Integration and novel selectors.

**Strengths:**

1) The fact that your method can be integrated into existing QSW/RQSW pipelines without changing downstream losses or gradients is a huge advantage. It means adoption is easy.

2) Using Bayesian Optimization  to drive projection selection is a very logical and promising direction. I would like to point out this solution

3) High-dimensional experiments

**Weaknesses:**

1) Comparison with other image-to-image models like Cycle GAN, DSBM or Optimal transport solvers

**Questions:**

1) This is the critical trade-off. How much more computational time does their BO-driven selection add compared to simpler random or linear selection methods? The definition of "modest" can be subjective, and the overhead might be acceptable for some applications but not others.

2) Could you provide expeirmtns with translation with resolution 128x128?

---

> ### Author Response · Authors · 2025-11-23
>
> We’re grateful to the reviewer for their positive evaluation of our work and for their constructive suggestions for improving our manuscript.  We’re pleased to address the reviewer’s comments in our revised draft (see OpenReview):
>
> W1: The reviewer notes several models used for image-to-image transfer tasks.  In our paper, we are focused on competing with existing SW methods—as our algorithms are inextricably rooted in SW—so we did not include comparisons to other non-SW models like CycleGAN.  However, the reviewer does mention optimal transport solvers, so we wanted to highlight that the QSW methods we compare against _are_ optimal transport solvers.  We also compare against a simple uniform Monte Carlo SW method (another optimal transport solver).  Together we hope that this at least mostly addresses this point, but if there is a specific comparison or experiment the reviewer would like to see, please just let us know.
>
> Q1: The reviewer asks an important question.  As noted in our revised Section 2, QMC-based methods like QSW require fewer samples than uniform MC-based SW—there is a big-O difference—and we expect our BO-based methods to usually perform at least as sample-efficiently as QSW.  However, despite being more sample-efficient, BO-based methods are more computationally expensive for each iteration/evaluation (see e.g. existing remarks in Section 3.1).  So the appropriate choice for a given application may indeed be a tradeoff between considerations like the cost of evaluating f versus the cost of evaluating the BO kernel.  To expand on our discussion of this topic in the paper, we have added a new Section 3.3 (“Computational Overhead”).
>
> Q2: We interpreted the reviewer’s question as referring to the image-to-image style transfer task (as in Section 4.4).  We have added experiments estimating SWD for two distributions that are separated by a translation, using L=10 and L=100, both for images with resolution 128 x 128.  The results are in Appendix A.5.  The results are quite positive for ARBOSW in particular, so we are grateful for the reviewer’s suggestion to add this example.
>
> Please feel free to let us know any other follow up questions or comments as you evaluate our revised work.

---

### Author Response · Authors · 2025-11-23

We want to express our gratitude again to each of the reviewers for the high-quality and constructive reviews across the board.  We are happy that from the initial scores, the most confident reviewers of our manuscript (5,4) were also the most positive (8,6).

As enumerated below in the responses to the individual reviewers, we have substantially improved our manuscript thanks to various suggestions in the reviews.  Highlights of our revised manuscript include:
- A substantially rewritten Background section (Section 2), expanded to cover QMC and BO methods
- Additional discussion in the conclusion relating our work to energy-based SW, as well as possible future extensions of our method using neural rather than GP surrogates
- Additional image-to-image transfer experiments demonstrating winning performance of ARBOSW (Appendix A.5)
- A new subsection discussing the computational overhead/complexity of BO-based SW methods (Section 3.3)
- Additional experiments demonstrating robustness of our methods against choice of kernel and acquisition parameter values (Appendix A.1)
- Added numerical results showing the deficiency of Bayesian quadrature (Appendix A.2)
- Added ablation study on our refresh parameter $R$ (Appendix C)
- Various meaningful improvements to several sentences and claims highlighted by reviewers

We are grateful for our significantly improved manuscript to be considered for sharing with the ICLR community.  We remain ready to address any further improvements suggested by the reviewers.

---

### Author Response · Authors · 2025-11-29

Hi all,

We are sharing one further comment in light of the unprecedented ICLR “leaks” that occurred. (Of course, we’d first like to note that we had no knowledge of nor participation in any leaks, and we are willing to sign an affidavit or provide any other guarantees to that end.)

Since reviewers can no longer reply, we are not sure if reviewer AKwQ would have updated their score, but we feel confident that we addressed all their concerns with our manuscript (and then some - given the improvements we made thanks to the other reviewers).

We would also like to highlight that, as is clear from the kind discussion below with reviewer EUMw, their score had been raised from negative to positive.  We understand revised scores are not being taken into account, but we thank the AC in advance for noting the positive feedback posted about our revised work.

We thank all the reviewers again for the productive discussion over the past couple weeks, which greatly helped us in producing an improved revised manuscript.

Best regards,

The Authors

---

### Meta-Review · Area_Chair_nzyK · 2025-12-28

**Summary:**

This paper proposes an efficient method for computing the sliced Wasserstein (SW) distance by adaptively learning projection directions using Bayesian Optimization (BO). Prior methods such as quasi-Monte Carlo (QMC) pseudo-uniformly sample projection directions, while this paper learns the adaptive weights from previously selected slices using a Gaussian process nonparametric regression formulation. Several BO methods are considered and designed as plug-and-chug into SW loss optimization. Empirical results with synthetic and real data are shown to outperform QSW baselines with reasonable runtime overhead.

The reviewers pointed out several merits of the proposed framework, including the novelty of combining BO and selection of SW projection directions; seamless integration into existing SW pipeline without changing loss function and gradients; comparison with benchmark baselines across different tasks; promising empirical evidence.

The reviewers also raised some concrete concerns, including the tradeoff between computation efficiency QSW and BO-based methods; biased estimation of BOSW; and more explicit clarification of the computational advantages over prior methods.

**Reviewer Concerns:**

In the rebuttal, the authors clarified the limitations of the high-dimensional scalability of BOSW and how to address the bias in estimating the SW. In the revised version, the authors added a good amount of numeric experiments to support their rebuttal points.

**Reviewer Scores:**

5 reviewers submitted their comments and scores (2/4/6/8/6) with confidence (3/3/2/5/4), with average score 5.2 and average confidence 3.4.

Based on the discussions between the authors and reviewers, multiple reviewers were satisfied with the response because of either clarified presentation or strengthened numeric experiments. Positive scores were maintained, and a negative score was indicated to increase. I expect if the discussions were continuing, the reviewers would have reached an overall positive consensus of this paper.

---

### Decision · Program_Chairs · 2026-01-26

Accept (Poster)